# Lipid biosynthesis enzyme Agpat5 in AgRP-neurons is required for insulin-induced hypoglycemia sensing and glucagon secretion

Anastasiya Strembitska[1], Gwenaël Labouèbe[1], Alexandre Picard[1], Xavier P. Berney[1], David Tarussio[1], Maxime Jan[2] & Bernard Thorens [1] ✉

The counterregulatory response to hypoglycemia that restores normal blood glucose levels is an essential physiological function. It is initiated, in large part, by incompletely characterized brain hypoglycemia sensing neurons that trigger the secretion of counterregulatory hormones, in particular glucagon, to stimulate hepatic glucose production. In a genetic screen of recombinant inbred BXD mice we previously identified *Agpat5* as a candidate regulator of hypoglycemia-induced glucagon secretion. Here, using genetic mouse models, we demonstrate that *Agpat5* expressed in agouti-related peptide neurons is required for their activation by hypoglycemia, for hypoglycemia-induced vagal nerve activity, and glucagon secretion. We find that inactivation of *Agpat5* leads to increased fatty acid oxidation and ATP production and that suppressing *Cpt1a*-dependent fatty acid import into mitochondria restores hypoglycemia sensing. Collectively, our data show that AgRP neurons are involved in the control of glucagon secretion and that *Agpat5*, by partitioning fatty acyl-CoAs away from mitochondrial fatty acid oxidation and ATP generation, ensures that the fall in intracellular ATP, which triggers neuronal firing, faithfully reflects changes in glycemia.

The defense against hypoglycemia is a major survival function. It is coordinated by a distributed glucose sensing system which comprises hypoglycemia-activated (glucose-inhibited or GI) neurons located in different brain nuclei and which control autonomic nervous activity, the hypothalamic-pituitary-adrenal axis, and, eventually, the secretion of the counterregulatory hormones, glucagon, epinephrine, nor-epinephrine, corticosterone and growth hormones[1–3]. These hormones reduce insulin secretion, decrease insulin-stimulated glucose uptake, and trigger hepatic glucose production to restore normoglycemia. In addition to this hormonal counterregulatory response (CRR), hypoglycemia also triggers search for, and ingestion of sugar-containing food in order to replenish the body glucose stores[1].

Hypoglycemia-activated GI neurons are present in the brainstem, including the dorsal vagal complex (DVC) and the parabrachial nucleus, and in several hypothalamic nuclei, including the para-ventricular (PVN), ventromedial (VMN), dorsomedial, and arcuate (ARC) nuclei[1,4]. In the hypothalamus, insulin-induced hypoglycemia (IIH) strongly activates neurons of the PVN and ARC[5,6] suggesting an important role of these nuclei in initiating the CRR. These nuclei are connected to the DVC, to control autonomic nervous system activation in response to IIH[7].

The mechanisms of hypoglycemia sensing that induce neuron firing are not completely characterized and appear to rely on different processes. For instance, in the VMN, GI neuron activation depends on

[1]Center for Integrative Genomics, University of Lausanne, 1015 Lausanne, Switzerland. [2]Bioinformatics Competence Center, University of Lausanne, 1015 Lausanne, Switzerland. ✉e-mail: Bernard.thorens@unil.ch

the energy sensor AMP-activated protein kinase (AMPK)[8], which is activated by a fall in intracellular energy level, and which regulates the activity of an ion channel, possibly the cystic fibrosis transmembrane conductance regulator (CFTR)[9]; AMPK also preserves GI neurons function by upregulating the expression of anti-oxidant proteins, in particular thioredoxin 2[8]. In agouti-related peptide (AgRP) neurons of the ARC, an alternate mechanism for hypoglycemia sensing is active[10,11]. This depends on the fall in intracellular ATP that accompanies developing hypoglycemia leading to reduced $Na^+/K^+$ATPase activity, plasma membrane depolarization and neuronal firing. In the ARC, but not in the VMN or lateral hypothalamus, the ATP level is indeed reduced during fasting and there is a linear relationship between extracellular glucose concentrations and intracellular ATP levels. Furthermore, intra-ARC injection of the $Na^+/K^+/$ATPase inhibitor ouabain, strongly increases food intake as a result of AgRP neuron activation[11]. Thus, fluctuations in intracellular ATP levels critically control AgRP neurons activity.

The secretion of counterregulatory hormones is mainly triggered by a fall in blood glucose concentration but is also under the regulatory influences of other metabolic signals. For instance, activation by hypoglycemia of parabrachial GI neurons, which stimulate glucagon secretion through their projections onto VMN neurons, is markedly decreased by leptin acting on the parabrachial neurons[12]. Thus, there is evidence that the overall CRR, although directed by glucose levels, is also under the control of signals of whole-body energy status. How hypoglycemia sensing is influenced by other metabolic signaling pathways is only poorly understood.

In a preceding study, we performed a genetic screen using a panel of recombinant inbred BXD mice to identify novel hypothalamic regulators of insulin-induced glucagon secretion[5]. We identified two QTLs, one on chromosome 8 and one on chromosome 15. Combined with transcriptomic analysis of the hypothalamus of the BXD mice, we identified *Irak4*, located in the QTL of chromosome 15, as a candidate gene whose expression negatively correlated with the glucagon trait. We confirmed that *Irak4*, by modulating hypothalamic Il-1β signaling, indeed controls hypoglycemia-induced glucagon secretion[5].

Here, we characterized the role of *Agpat5*, the candidate gene of the chromosome 8 QTL, in glucagon secretion. Agpat5 (1-acylglycerol-3-phosphate-O-acyltransferase 5), one of five Agpat isoforms, is associated with the mitochondrial membrane and catalyzes the production of phosphatidic acid (PA) from lysophosphatidic acid (LPA) and acyl-CoAs[13]. We demonstrated that *Agpat5* inactivation in AgRP neurons led to reduced hypoglycemia-induced glucagon secretion. This was associated with reduced activation by hypoglycemia of AgRP neurons as assessed by c-Fos immunostaining, patch clamp analysis, and in vivo fiber photometry; and with defective activation of the vagal nerve. We further demonstrate that *Agpat5* inactivation increased *Cpt1a*-dependent mitochondrial fatty acid β-oxidation (FAO), oxygen consumption rate (OCR), and ATP production. These data show that *Agpat5* is required to divert fatty acyl-CoAs from entering the mitochondria to prevent FAO-dependent ATP production, thereby ensuring proper hypoglycemia sensing.

## Results

### *Agpat5* inactivation in AgRP neurons impairs glucagon secretion

A previous genetic screen for hypoglycemia-induced glucagon secretion using a panel of 36 recombinant inbred BXD mouse lines identified a QTL on chromosome 8[5]. Analysis of RNASeq data from the hypothalamus of the same panel of BXD mice, revealed that expression of only three genes located in this QTL displayed significant correlation with the glucagon trait, *Agpat5* and two genes of unknown function (*261005L07Rik* and *LOC547150*). Here, we further analyzed the hypothalamic RNASeq data from the BXD mice and identified a *cis*-eQTL for *Agpat5* expression that was located in the same genomic interval of chromosome 8 that controls glucagon secretion, between

markers rs13479628 and rs33450716 (Fig. 1A, B). We then determined that the plasma glucagon levels correlated, across the BXD lines, with *Agpat5* mRNA expression in the hypothalamus ($p = 0.0022$, $r = 0.493$) (Fig. 1C), but not in the liver ($p = 0.1192$, $r = 0.2644$) (Fig. 1D) nor in the subcutaneous adipose tissue ($p = 0.5495$, $r = -0.1080$) (Fig. 1E). Thus, these genetic and transcriptomic data indicated a role for hypothalamic *Agpat5* expression in the regulation of glucagon secretion.

*Agpat5* is ubiquitously expressed[14] and in situ hybridization confirmed that it was present in all hypothalamic nuclei, including in the ARC and PVN (Fig. 1F). As insulin-induced hypoglycemia strongly activates PVN neurons and AgRP neurons of the ARC[5,6] (see below) we decided to assess the impact of inactivating *Agpat5* in the PVN and in AgRP neurons on hypoglycemia-induced glucagon secretion. We generated *Agpat5^flox/flox^* mice (Fig. 1G, H) and crossed them first with *Sim1-Cre* mice to inactivate *Agpat5* in the PVN (*Sim1^Agpat5KO^* mice) (Supplementary Fig. 1A). Male and female *Agpat5^flox/flox^* and *Sim1^Agpat5KO^* mice displayed identical body weight (Supplementary Fig. 1B), glucose tolerance (Supplementary Fig. 1C, D), insulin sensitivity (Supplementary Fig. 1E, F), and hypoglycemia-stimulated glucagon secretion (Supplementary Fig. 1G–J).

We next inactivated *Agpat5* in AgRP neurons by crossing *Agpat5^flox/flox^* mice with *AgRP-Cre* mice (*AgRP^Agpat5KO^* mice). We confirmed that recombination took place only in the ARC (Fig. 1I) and this did not change the number of AgRP neurons, as determined by fluorescence microscopy quantification of tdTomato-positive cells in the hypothalamus of *AgRP^Cre/+^;Rosa26^tdTom^* and *AgRP^Agpat5KO^;Rosa26^tdTom^* mice (Fig. 1J, K). *Agpat5^flox/flox^* and *AgRP^Agpat5KO^* mice of both sexes showed no difference in body weight gain, in glucose tolerance tests nor in insulin tolerance tests, except for a significant delay in the glycemia recovery rate in female mice (Supplementary Fig. 2A–F). The identical rates of hypoglycemia development after insulin injection in control and mutated mice (Supplementary Fig. 2E, F) also indicated that insulin sensitivity was not modified by *Agpat5* inactivation in AgRP neurons. Male *AgRP^Agpat5KO^* mice had normal feeding patterns as measured over a 24 h *ad libitum* feeding period (Supplementary Fig. 2G). Moreover, there was no significant difference in food consumption over a 4 h refeeding period following a 16 h fast (Supplementary Fig. 2H).

We then tested the glucagon response to IIH. Mice were first injected intraperitoneally (i.p.) with saline and plasma glucagon was measured one hour after the injection. The experiment was repeated two weeks later with i.p. injection of insulin. Insulin induced the same level of hypoglycemia in *Agpat5^flox/flox^* and *AgRP^Agpat5KO^* mice of both sexes (Fig. 2A, B). However, plasma glucagon levels were significantly less increased in male and female *AgRP^Agpat5KO^* mice as compared to *Agpat5^flox/flox^* mice (Fig. 2C, D). We then performed hyperinsulinemic-hypoglycemic clamps to precisely control the level of hypoglycemia and measured the rate of glucose infusion required to maintain hypoglycemia, and the plasma glucagon concentrations at the end of the clamps. *Agpat5^flox/flox^* and *AgRP^Agpat5KO^* mice reached the exact same hypoglycemic levels (Fig. 2E) but the glucose infusion rate was higher in the mutated mice (Fig. 2F, G) and these mice secreted less glucagon (Fig. 2H). Thus, *Agpat5* in AgRP neurons was required for normal hypoglycemia-induced glucagon secretion.

### *Agpat5* inactivation reduces the number of glucose responsive AgRP neurons and vagal activity

To determine whether *Agpat5* was affecting the glucose sensitivity of AgRP neurons, we first quantitated c-Fos expression by immunofluorescence microscopy after saline or insulin injections. Insulin-induced hypoglycemia reached the same level in control and mutated mice (Fig. 3A). The number of c-Fos positive cells markedly increased after insulin injection in the ARC of *Agpat5^flox/flox^* mice, but not in *AgRP^Agpat5KO^* mice (Fig. 3B, C).

Next, we assessed the glucose responsiveness of AgRP neurons by patch clamp analysis. Acute brain slices were prepared

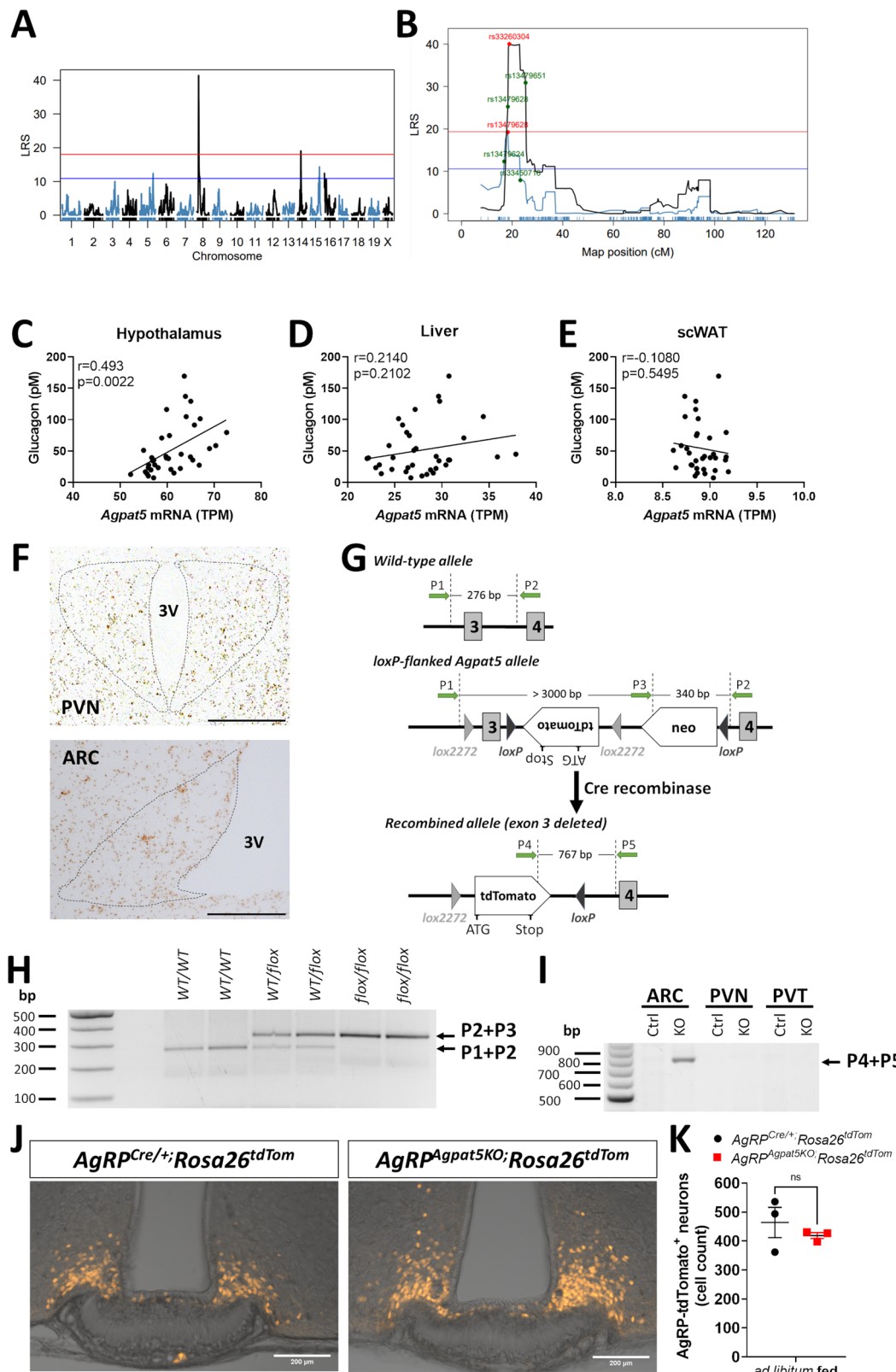

from *AgRP^{Cre/+};Rosa26^{tdTom}* mice and *AgRP^{Agpat5KO};Rosa26^{tdTom}* mice and patch clamp recordings of AgRP neurons were performed while switching the extracellular glucose concentrations from 5 mM to 0.1 mM. In *AgRP^{Cre/+};Rosa26^{tdTom}* mice 72 percent of AgRP neurons were GI neurons and the others 28 percent were glucose non-responder (NR). In *AgRP^{Agpat5KO};Rosa26^{tdTom}* mice, the percentage of AgRP GI neurons was reduced to 33 percent of the

recorded neurons and NR neurons increased to 67 percent (Fig. 3D, E). The membrane potential of the GI neurons measured in both groups of mice were similarly increased by lowering the glucose concentration from 5 to 0.1 mM glucose (Fig. 3F). Although the GI neurons membrane resistance increased upon exposure to low glucose concentrations in *AgRP^{Cre/+};Rosa26^{tdTom}* mice, no statistically significant difference was observed in

**Fig. 1 | Hypothalamic *Agpat5* and insulin-induced hypoglycemia.** Hypothalamic *Agpat5* on chromosome 8 is a candidate regulator of insulin-induced hypoglycemia. **A** QTL mapping of hypothalamic *Agpat5* expression reveals a *cis*-eQTL on chromosome 8. **B** The *cis*-eQTL for *Agpat5* (black) overlaps with the cQTL for glucagon secretion (blue). **C** Correlation between *Agpat5* mRNA expression in the hypothalamus of the BXD mice and the glucagon trait. **D, E** No correlation was found between *Agpat5* mRNA expression in the liver (**C**) or adipose tissue (**D**) of the BXD mice and the glucagon trait. **F** Representative image of in situ hybridization detection of *Agpat5* mRNA in the PVN *(top panel)* and ARC *(bottom panel)* hypothalamic nuclei. Scale bar, 200 μm. **G** Structure of the wild-type (WT) *Agpat5* allele around exons 3 and 4 *(top)*, of the *flox* allele *(middle)* and of the recombined allele

*(bottom)*. The primers used for genotyping (P1, P2, P3, P4, P5 in green), their location and size of the amplified fragments are indicated. **H** Genotyping of *WT* and *flox* alleles using ear DNA. **I** PCR analysis of the recombined allele in DNA from microdissected ARC, PVN and paraventricular thalamus (PVT) of *Agpat5flox/flox* (Ctrl) and KO *AgRPAgpat5KO* (KO) mice. H, I Representative images are presented. **J** tdTomato expression in the ARC of *AgRPCre/+;Rosa26tdTom* and *AgRPAgpat5KO;Rosa26tdTom* mice. Scale bars, 200 μm. **K** Quantification of AgRP neurons in the ARC of *AgRPCre/+;Rosa26tdTom* and *AgRPAgpat5KO;Rosa26tdTom* mice. *n* = 3 mice per genotype, three different sections/mouse. Data are mean ± SEM. Two-tailed, unpaired Student's test.

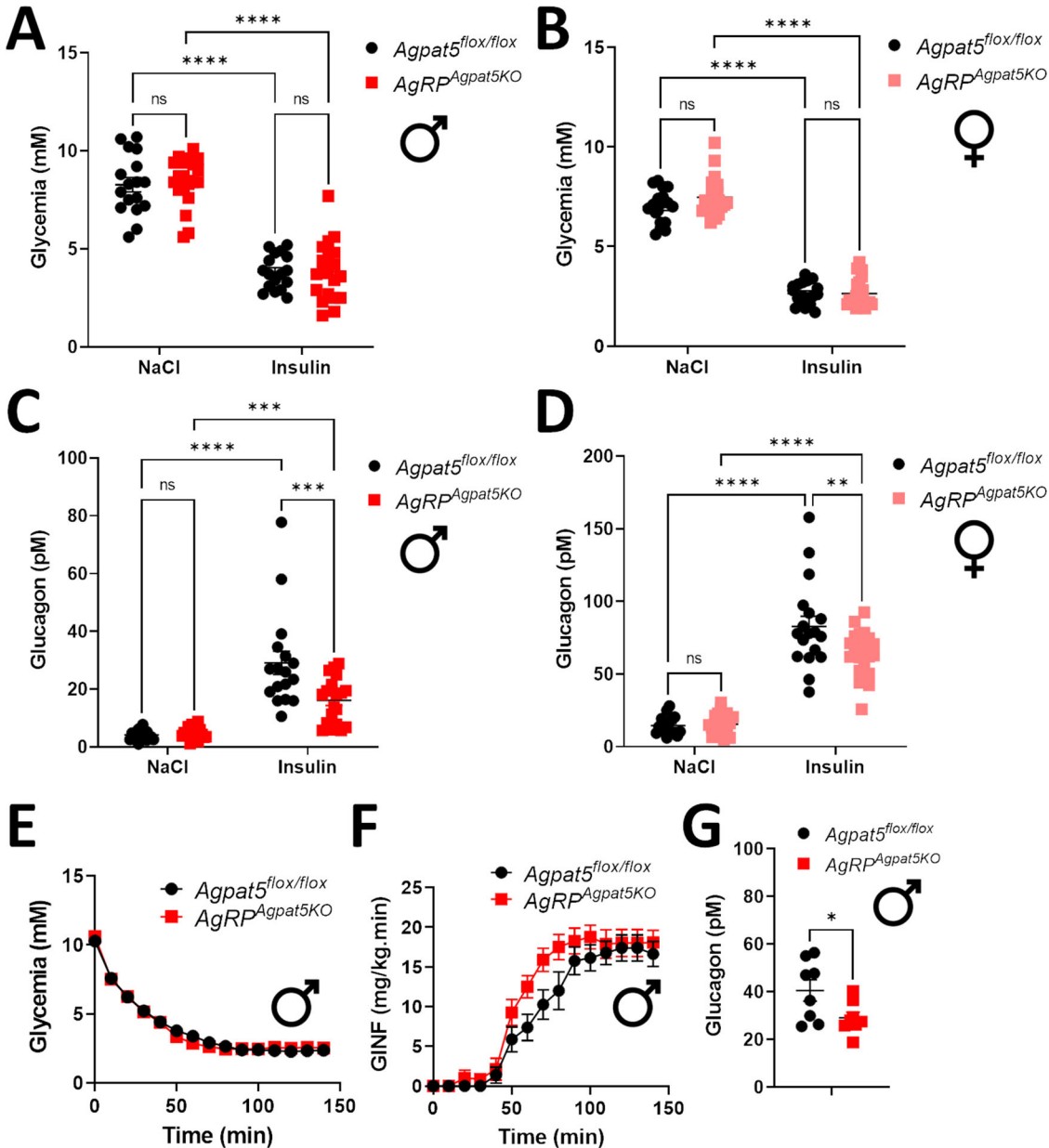

**Fig. 2 | *AgRPAgpat5KO* mice display reduced hypoglycemia-induced glucagon secretion. A, B** Glycemia in *Agpat5flox/flox* mice and *AgRPAgpat5KO* mice one hour after NaCl or insulin injection; **A** male mice, **B** female mice. Plasma glucagon levels one hour after NaCl or insulin injection in *Agpat5flox/flox* and *AgRPAgpat5KO* mice **C** male mice, **D** female mice. **A–D** Two independent cohorts, *n* = 18–22 mice per genotype. Data are mean ± SEM, two-way ANOVA, *** *p* < 0.001 and ****\*p* < 0.0001, two-way ANOVA

with Tukey's multiple comparisons correction. **E–H** Hyperinsulinemic-hypoglycemic clamps performed in *Agpat5flox/flox* mice and *AgRPAgpat5KO* male mice: **E** Glycemic levels. **F** Glucose infusion rates. Data are mean ± SEM, (repeated measurements) with Sidak's multiple comparisons correction. **G** Plasma glucagon levels at the end of the clamp. Data are mean ± SEM, *n* = 8 mice. \**p* < 0.05, two-tailed, unpaired Student's t-test.

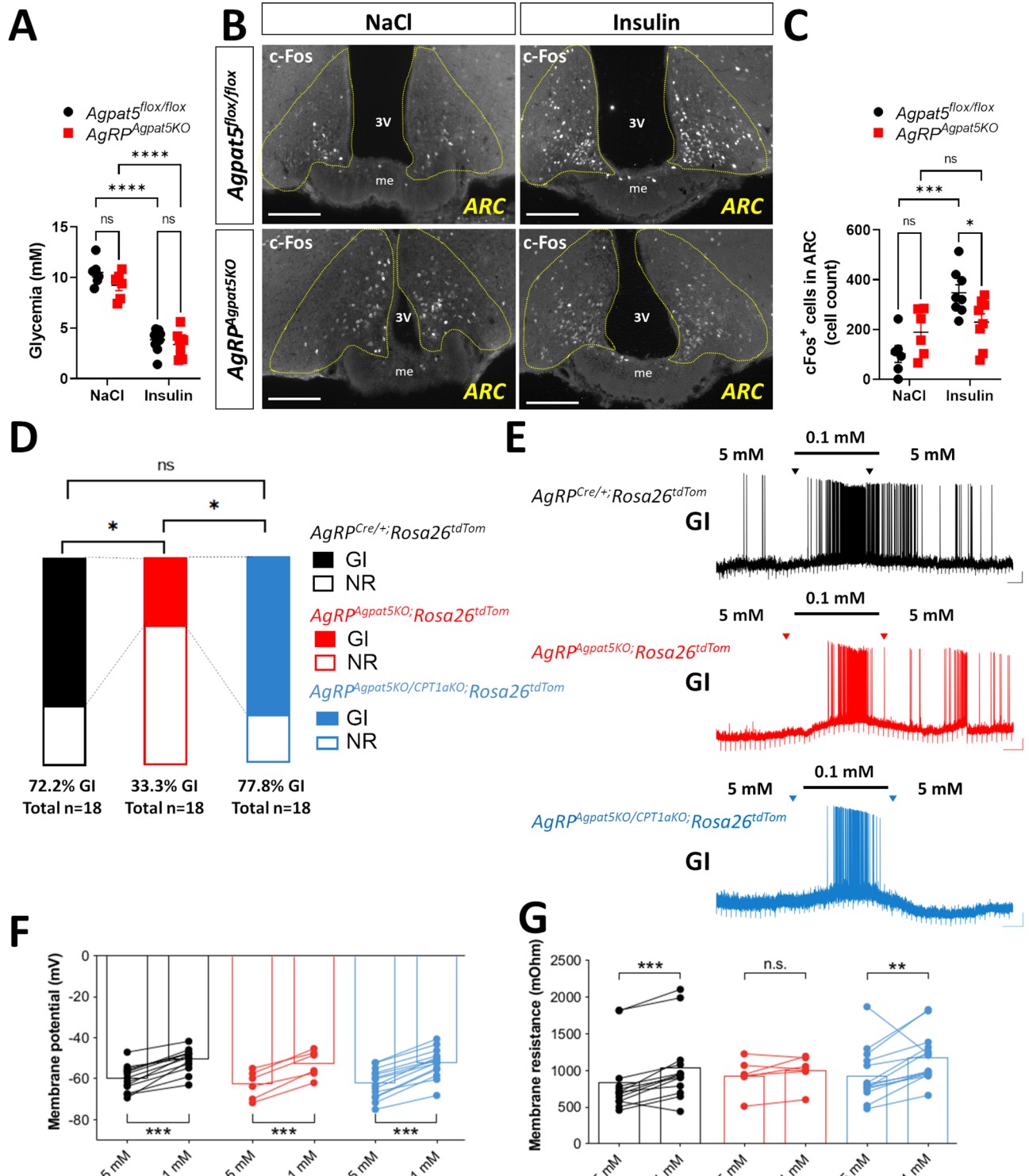

**Fig. 3 | Impaired activation by hypoglycemia of AgRP neurons in *AgRP^AgpatSKO* mice. A**–**C** *Agpat5^flox/flox* and *AgRP^AgpatSKO* male mice were injected i.p. with NaCl or insulin and their brains were collected one hour later for c-Fos immunodetection. **A** Glycemic levels after NaCl or insulin injections. **B** c-Fos immunostaining. Scale bar, 200 μm. **C** Quantification of c-Fos staining in the ARC. Data are mean ± SEM. c-Fos positive nuclei have been counted at three different bregmas/mouse. Two independent mouse cohorts, $n = 6$–7 mice per group. Data are mean ± SEM, \*\*\*$p < 0.001$ and \*\*\*\*$p < 0.0001$, two-way ANOVA with Tukey's multiple comparisons correction. The glucose responsiveness of AgRP neurons of mice of the indicated genotypes was analyzed by patch clamp analysis performed on acute brain slices.

AgRP neurons were identified by the expression of tdTomato. **D** Proportions of GI and NR neurons in mice from each genotype (*AgRP^Cre/o+; Rosa26^tdTom* vs *AgRP^AgpatSKO;Rosa26^tdTom*). \*$p < 0.05$, two-sided Fisher's exact test. **E** Example of GI neurons recorded in the three indicated mouse lines. The concentration of glucose in the superfusates is indicated at the top of each recording. **F** Change in membrane potential induced by 0.1 mM glucose in AgRP GI neurons from each mouse genotype. **G** Change in membrane resistance. Data from $n = 18$ neurons/$n = 8$–10 mice per genotype. Before-after graphs display individual values. \*\*\*$p < 0.001$, two-tailed, paired Student's t-test.

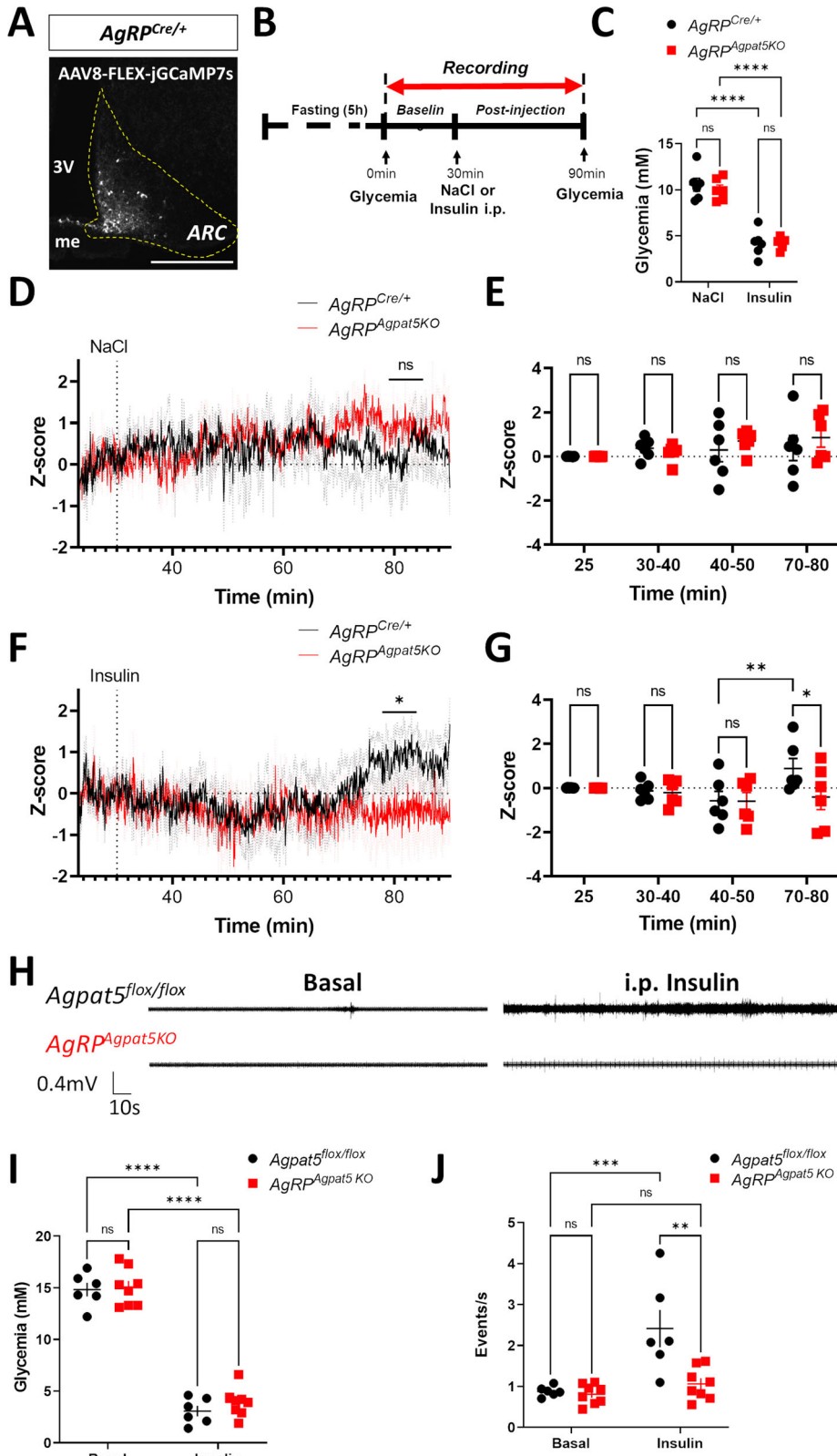

AgRP$^{Agpat5KO}$;Rosa26$^{tdTom}$ mice, possibly due to the small number of GI neurons detected in this genotype (Fig. 2G). Thus, inactivation of Agpat5 in AgRP neurons suppressed the GI response of a subset of these neurons but left unaffected the response of the remaining GI neurons.

To determine whether the glucose responsiveness of AgRP neurons was affected by Agpat5 inactivation in living mice, we performed fiber photometry recordings. AgRP$^{Cre/+}$ mice and AgRP$^{Agpat5KO}$ littermates received stereotactic injection of an AAV9-syn-FLEX-jGCamp7s-WPRE in the ARC and were implanted at the same site with an optic fiber. GCaMP7 expression in the ARC was verified by fluorescence microscopy at the end of the experiments (Fig. 4A). On the day of the experiments, mice were fasted for 5 h before initiating GCaMP7 fluorescence recording for a 30 min basal period followed by an i.p.

**Fig. 4 | *Agpat5* inactivation blunts hypoglycemia-induced AgRP neuron activity and vagal nerve firing. A** Expression of GCaMP7 in the ARC of *AgRP^Cre/+^* mice following the stereotactic injection of AAV-FLEX-GCaMP7. Scale bar, 200 µm. **B** Scheme of the fiber photometry experiment. **C** Glycemia following NaCl or insulin injection in *AgRP^Cre/+^* and *AgRP^AgpatSKO^* mice, *n* = 6. Data are mean ± SEM, ****\*p* < 0.0001, two-way ANOVA with Tukey's multiple comparisons correction. **D** Recordings of the GCaMP7 signal in AgRP neurons of *AgRP^Cre/+^* and *AgRP^AgpatKO^* mice following NaCl injection. Data are mean ± SEM. **E** Quantification over 10 min periods of the GCaMP7 signal of (**D**). Data are mean ± SEM, two-way ANOVA with Sidak's multiple comparisons correction. **F** Recordings of the GCaMP7 signal following insulin injection. Data are mean ± SEM. **G** Quantification over 10 min periods of the GCaMP7 signal of (**F**). Data are mean ± SEM, *\*p* < 0.05, two-way ANOVA with Sidak's multiple comparisons correction. Data are from three independent experiments, *n* = 6 mice. **H** Representative traces of parasympathetic nerve firing rate in *Agpat5^flox/flox^* (top) and *AgRP^AgpatSKO^* (bottom) mice in the basal state (left) and following insulin injection (right). **I** Glycemic levels of *Agpat5^flox/flox^* mice and *AgRP^AgpatSKO^* mice in the basal state and 1 h following insulin injection. **J** Quantification of vagal nerve firing in the basal state and following insulin-induced hypoglycemia, *n* = 6–8 mice. **I, J** Data are mean ± SEM, *\*\*p* < 0.01, *\*\*\*p* < 0.001 and *\*\*\*\*p* < 0.0001, two-way ANOVA with Tukey's multiple comparisons correction.

injection of saline or insulin and a further 60 min recording period (Fig. 4B). Insulin induced the same level of hypoglycemia in both groups of mice (Fig. 4C). The GCaMP7 fluorescence signal after saline injection remained the same during the recording period in *AgRP^Cre/+^* and *AgRP^AgpatSKO^* mice (Fig. 4D), as quantitated over 10 min periods (Fig. 4E). After insulin injection the GCaMP7 signal increased in *AgRP^Cre/+^* mice with a lag period of ~40 min; this increase was, however, not observed in *AgRP^AgpatSKO^* mice (Fig. 4F, G). Thus, inactivation of *Agpat5* led to a reduction in the number of AgRP neurons that can be activated by hypoglycemia as detected by c-Fos staining, patch clamp analysis, and in vivo intracellular Ca²⁺ recordings.

As vagal nerve activation during hypoglycemia stimulates glucagon secretion[15], we recorded vagal nerve activity in *Agpat5^flox/flox^* and *AgRP^AgpatSKO^* littermates under basal conditions for 30 min and for one hour after i.p. injection of insulin (Fig. 4H). The glycemic levels upon insulin injection were similar between the two groups (Fig. 4I). Vagal activity increased 2-fold in *Agpat5^flox/flox^* mice ~40–50 min after insulin injection, but not in *AgRP^AgpatSKO^* animals (Fig. 4J). Thus, reduced AgRP activation by hypoglycemia in *AgRP^AgpatSKO^* mice led to impaired vagal nerve firing and reduced glucagon secretion.

## *Agpat5* controls FAO and ATP production

We next investigated how *Agpat5* could interfere with glucose sensing by AgRP neurons. Agpat5 is associated with mitochondria and catalyzes the production of PA from LPA and fatty acyl-CoAs[13]. PA has been described as a competitive inhibitor of dynamin-related protein-1 (Drp1), the main regulator of mitochondrial fission, preventing mitochondrial network fragmentation[16,17]. As mitochondrial dynamics in AgRP and POMC neurons has previously been reported to control their glucose responsiveness[18–20] we first assessed whether *Agpat5* inactivation could lead to changes in mitochondrial morphology. Electron microscopy analysis of mitochondria in AgRP neurons of *AgRP^Cre/o+^;Rosa26^tdTom^* and *AgRP^AgpatSKO^;Rosa26^tdTom^* male mice (Fig. 5A) revealed that mitochondrial area and perimeter cumulative distributions were identical in both groups of mice (Fig. 5A–C). Thus, a role for *Agpat5* in controlling mitochondrial dynamics was not supported.

Activation of AgRP neuron firing by hypoglycemia is triggered by a fall in intracellular ATP content that reduces the activity of the Na⁺/K⁺ATPase, leading to membrane depolarization[10,11]. Thus, in conditions of hypoglycemia, the role of Agpat5 may be to prevent fatty acyl-CoAs from entering the mitochondria for FAO ensuring that reduced ATP production reflects the decrease in blood glucose concentrations. To test this hypothesis, we evaluated whether silencing *Agpat5* expression in the GT1-7 neuronal cell line would regulate mitochondrial oxygen consumption rate and ATP production. GT1-7 cells were transfected with two different *Agpat5*-specific siRNAs or a control siRNA. The efficacy of *Agpat5* silencing was tested by western blot analysis using a newly produced and validated Agpat5 polyclonal antibody (Supplementary Fig. 3). Agpat5 was detected in GT1-7 cells using an immunoprecipitation/immunoblotting protocol (Fig. 5D); *Gfp*- and *Agpat5-Flag*-transfected HEK293T cells served as negative and positive controls, respectively (Fig. 5D). Silencing *Agpat5* expression by

both siRNAs led to a 40–50 percent reduction in Agpat5 protein expression (Fig. 5E).

Oxygen consumption rates (OCR) measured by Seahorse analysis showed that in GT1-7 cells, mitochondrial respiration did not decrease with glucose supplementation (Fig. 5F), indicating that GT1-7 cells preferentially use mitochondrial respiration and not glycolysis for ATP generation. Silencing *Agpat5* expression increased basal mitochondrial respiration and ATP production in GT1-7 cells (Fig. 5F,G). The results were consistent for both specific siRNAs, minimizing the possibility of an off-target effect (Fig. 5H).

We next tested the impact on OCR of inhibiting Cpt1 with etomoxir. Etomoxir had no effect on OCR in control siRNA-transfected cells, indicating that endogenous FAO rates are low in control GT1-7 cells (Fig. 5F, G). In contrast, etomoxir suppressed the increase in OCR and ATP production in *Agpat5*-silenced cells (Fig. 5F, G), indicating that *Agpat5* normally prevents increased FAO. To confirm this observation, we transfected GT1-7 cells with a control, an *Agpat5*-specific or a *Cpt1a*-specific siRNA. *Agpat5* and *Cpt1a* mRNAs expression were decreased by ≥50 percent by their respective siRNAs (Fig. 5K). Reducing *Cpt1a* expression had no effect on GT1-7 cells OCR. However, reducing *Cpt1a* expression abolished the increase in OCR and ATP production induced by *Agpat5* silencing (Fig. 5I, J). These results support the hypothesis that the role of *Agpat5* is to divert fatty-acyl-CoAs from *Cpt1*-mediated entry into mitochondria.

## Inactivation of *Cpt1a* in *AgRP^AgpatSKO^* mice restores glucose sensing

To determine whether the reduced glucose responsiveness of AgRP neurons from *AgRP^AgpatSKO^* mice could be restored by blocking FAO, we generated *AgRP^AgpatSKO;Cpt1aKO^* mice by crossing the *AgRP^AgpatSKO^* mice with *Cpt1a^flox/flox^* mice[21]. We first analyzed c-Fos expression in AgRP neurons of *Agpat5^flox/flox^;Cpt1a^flox/flox^* mice, *AgRP^AgpatSKO^* mice, and *AgRP^AgpatSKO;Cpt1aKO^* mice one hour after insulin injection. Mice from the three genotypes showed similar glycemia before and after the insulin injections (Fig. 6B). Quantification of the number of c-Fos positive cells indicated that hypoglycemia induced a lower number of c-Fos positive cells in *AgRP^AgpatSKO^* mice than in *Agpat5^flox/flox^;Cpt1a^flox/flox^* mice (Fig. 6A, and compared with Fig. 3B). Importantly, *AgRP^AgpatSKO;Cpt1aKO^* mice displayed the same number of c-Fos positive AgRP neurons as *Agpat5^flox/flox^;Cpt1a^flox/flox^* mice (Fig. 6 A, C).

We next tested whether *Cpt1a* inactivation in AgRP neurons affected their glucose responsiveness and glucose homeostasis. We first performed patch clamp analysis of AgRP neurons in *AgRP^Cre/+^;Rosa26^tdTom^* mice and *AgRP^Cpt1aKO^;Rosa26^tdTom^* male littermates. In both groups of mice the percentage of AgRP GI neurons was the same (~70 percent) (Supplementary Fig. 4A–C), and the same as observed in non-littermate *AgRP^Cre/+^;Rosa26^tdTom^* mice (see Fig. 3D). Changes in membrane potential and membrane resistance in response to switching glucose concentrations from 5 to 0.1 mM were also identical between both groups of mice (Supplementary Fig. 4D, E). Furthermore, *Cpt1a^flox/flox^* mice and *AgRP^Cpt1aKO^* littermates displayed identical glucose tolerance, insulin sensitivity, and glucagon response to insulin-induced hypoglycemia (Supplementary Fig. 4F–I). Thus, inactivating *Cpt1a* in

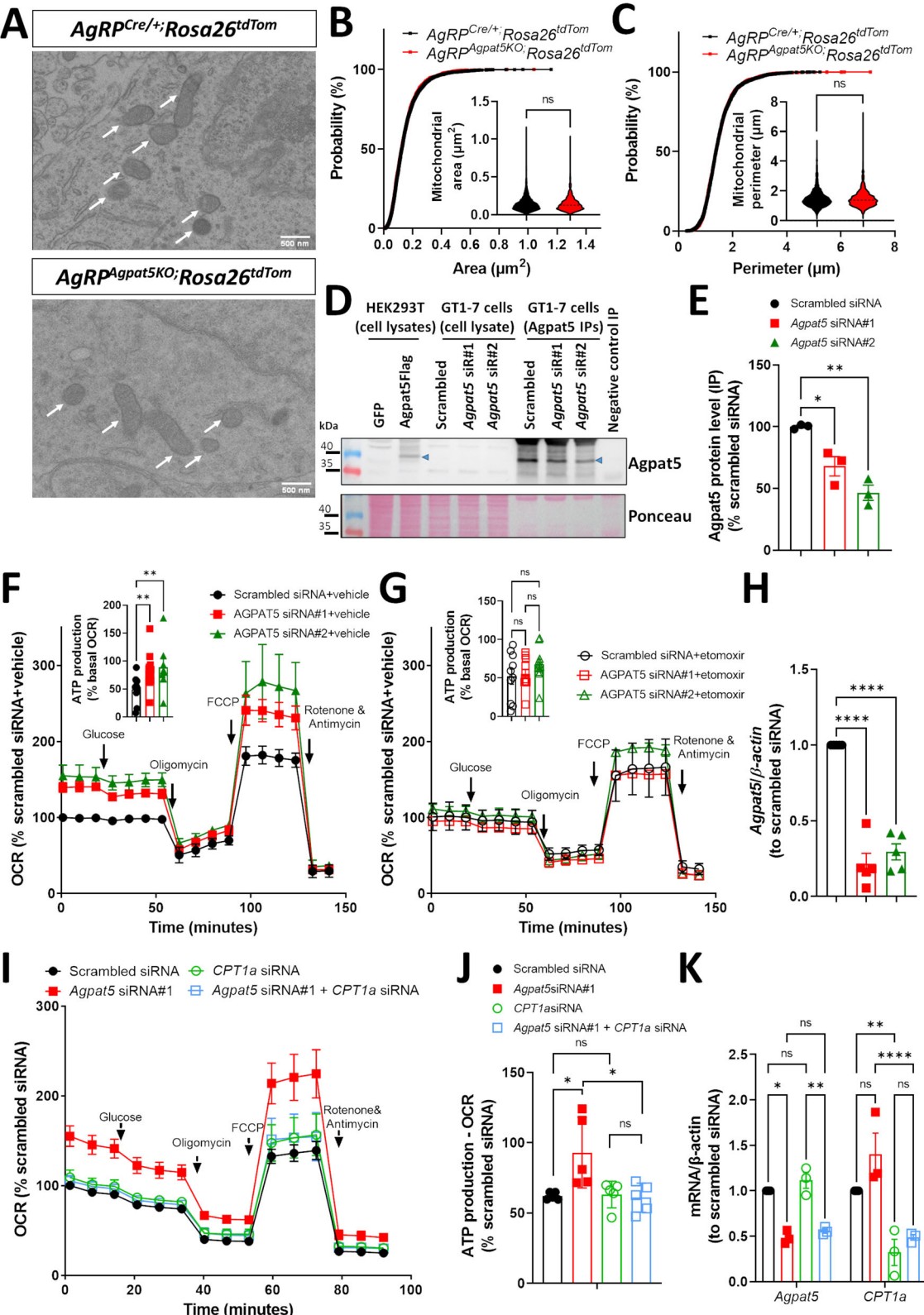

AgRP neurons did not impact their glucose responsiveness nor any of the measured glucose homeostasis parameters. Importantly, however, inactivation of *Cpt1a* in AgRP neurons lacking *Agpat5* (AgRP*Agpat5KO;Cpt1aKO;Rosa26tdTom* mice) restored the number of GI neurons found in control mice (Fig. 3D), which displayed similar changes in membrane potential and membrane resistance induced by low glucose concentrations.

We next assessed insulin-induced glucagon secretion. Mice from the three genotypes had the same body weight, the same basal glycemia and insulin-induced hypoglycemia (Fig. 6D, E). Glucagon secretion was significantly lower in *AgRPAgpat5KO* mice than in *Agpat5flox/flox;Cpt1aflox/flox* mice. In mice with inactivation of both *Agpat5* and *Cpt1a*, hypoglycemia did not induce a significant reduction in glucagon secretion as compared to *Agpat5flox/flox;Cpt1aflox/flox* mice (Fig. 6F).

**Fig. 5 | *Agpat5* inactivation does not modify mitochondria dynamics but increases ATP production.** Mitochondria morphology was analyzed in AgRP neurons of *AgRP^Cre/+;Rosa26^tdTom* mice and of *AgRP^AgpatKO;Rosa26^tdTom* littermate mice in *ad libitum* fed state. **A** Electron microscopy pictures of AgRP neurons of *AgRP^Cre/+; Rosa26^tdTom* mice and *AgRP^AgpatKO;Rosa26^tdTom* mice. AgRP neurons were identified by correlative light and electron microscopy. **B** Cumulative distribution of mitochondria areas. **C** Cumulative distribution of mitochondria perimeters, 3 mice per genotype; 90 AgRP neurons; 2424–2491 mitochondria. Scale bar, 500 nm. Two-tailed, paired Student's test. Western blot analysis of Agpat5 expression in GT1-7 cells transfected with a scrambled or two different *Agpat5*-specific siRNAs. **D** Detection of Agpat5 by immunoprecipitation followed by immunoblotting with the Agpat5 antibody. Lysates of HEK293T cells transfected with a *GFP* or an *Agpat5GFP* expression plasmid served as negative and positive controls for Agpat5, respectively. **E** Quantification of Agpat5 expression. $n = 3$ replicates. Data are mean ± SEM, $*p < 0.05$, $**p < 0.01$, one-way ANOVA with Dunnett's multiple comparisons correction. Oxygen consumption rates (OCR) and ATP production

(defined by difference between basal OCR and OCR after blocking ATP synthase with oligomycin) were measured in GT1-7 cells transfected with scrambled or *Agpat5*-specific siRNAs. The role of β-oxidation in ATP production was verified using the CPT1a inhibitor etomoxir or by siRNA-mediated *Cpt1a* silencing, $n = 9$–10 biological replicates. Data are mean ± SEM, $*p < 0.05$, one-way ANOVA with Tukey's post hoc test. **F** OCR in GT1-7 cells in the absence of etomoxir and quantification of ATP production by cells. **G** OCR in GT1-7 cells in the presence of etomoxir and quantification of ATP production by cells. **H** PCR quantification of *Agpat5* expression, $n = 5$ replicates. Data are mean ± SEM, $****p < 0.0001$, one-way ANOVA with Dunnett's multiple comparisons correction. **I** OCR in GT1-7 cells transfected with the scrambled siRNA, an *Agpat5* siRNA, and/or a *Cpt1a* siRNA. **J** Quantification of ATP production by cells in (**I**), $n = 5$ biological replicates, $n = 5$ technical replicates per assay. **K** PCR quantification of *Agpat5* and *Cpt1a* expression in cells used in (**J**), $n = 3$ biological replicates. **J**, **K** Data are mean ± SEM, $*p < 0.05$, $**p < 0.01$, $***p < 0.001$ and $****p < 0.0001$, two-way ANOVA with Tukey's multiple comparisons correction.

Together the above data support the hypothesis that the role of Agpat5 is to divert fatty-acyl-CoAs from entering the mitochondria to prevent an increase in ATP production that would impede neuronal firing upon hypoglycemia. A corollary of this hypothesis is that insulin-induced glucagon secretion in *AgRP^AgpatSKO* mice should be increased if circulating free fatty acids (FFAs) could be lowered. To verify this hypothesis, we treated *Agpat5^flox/flox* and *AgRP^AgpatSKO* mice with nicotinic acid (NA) to block lipolysis and reduce circulating FFAs. Plasma FFAs in 6 h fasted *Agpat5^flox/flox* and *AgRP^AgpatSKO* male mice were the same (224.8 ± 16.5 and 228.9 ± 23.2 μM, respectively) and were similarly reduced 30 min after NA injection (133.6 ± 5.3 and 138.4 ± 16.6 μM, respectively) (Fig. 6G). Thirty minutes after saline or NA injection, the mice received an i.p. injection of insulin. The hypoglycemic levels reached one hour later did not differ between of *Agpat5^flox/flox* and *AgRP^AgpatSKO* mice (Fig. 6H). Plasma glucagon levels in *Agpat5^flox/flox* mice and *AgRP^AgpatSKO* mice were identical after saline or NA injections (Fig. 6I). After insulin injection plasma glucagon was less increased in *AgRP^AgpatSKO* mice as compared to *Agpat5^flox/flox* mice, as expected. However, insulin-induced hypoglycemia after NA pre-treatment led to the same plasma glucagon levels in *Agpat5^flox/flox* mice and *AgRP^AgpatSKO* mice (Fig. 6I). Thus, lowering circulating FFA levels corrected the glucagon secretion defect of *AgRP^AgpatSKO* mice, in agreement with the hypothesis that reducing FAO and ATP production is required for normal hypoglycemia-induced activation of AgRP neurons and glucagon secretion.

## Discussion

Here, we investigated the role of *Agpat5* as a regulator of hypoglycemia-induced glucagon secretion. Our physiological studies show that inactivation of *Agpat5* in AgRP neurons suppresses their activation by hypoglycemia, impairs hypoglycemia-induced vagal nerve activity, and reduces glucagon secretion. Experiments performed in a hypothalamic cell line and in mice led to the conclusion that the role of *Agpat5* is to divert free fatty acids away from *Cpt1a*-dependent entry in the mitochondria, FAO, and ATP production. This ensures that a fall in glycemia leads to a commensurate decrease in intracellular ATP levels and AgRP neurons firing. The role of Agpat5 is, thus, to ensure that AgRP neurons respond faithfully to developing hypoglycemia.

In a previous screen of recombinant inbred BXD mice we identified two QTLs for insulin-induced glucagon secretion, one on chromosome 8 and the other on chromosome 15[5]. *Irak4*, located in the QTL of chromosome 15 has been confirmed as a genetically controlled, hypothalamic regulator of glucagon secretion. We found that increased *Irak4* expression, leading to increased Il-1ß signaling, reduced hypoglycemia-induced glucagon secretion[5].

*Agpat5* appeared as the most likely candidate regulator of hypoglycemia-induced glucagon secretion of the chromosome 8 QTL[5].

Interestingly, *Agpat5* has also been identified in another genetic screen of BXD mice as a determinant of basal insulinemia and whole-body insulin resistance[22] and cis-eQTLs were identified for liver and fat *Agpat5* expression. In addition, *Agpat5* silencing in these two tissues by antisense oligonucleotide treatment reduced basal insulinemia and improved glucose tolerance[22]. As our own screen was based on insulin-induced hypoglycemia *Agpat5* could, in principle, mediate its effect on glucagon secretion by controlling insulin sensitivity and hypoglycemia development and/or on hypoglycemia sensing. A correlation between the glucagon trait and *Agpat5* levels was, however, found only for *Agpat5* expression in the hypothalamus and not in the liver or fat. In addition, inactivation of *Agpat5* in AgRP neurons had no impact on basal insulinemia nor on insulin sensitivity as assessed in insulin tolerance tests. Furthermore, hypoglycemia induced by i.p. insulin injection or during a hypoglycemic clamp led to lower glucagon secretion in *AgRP^AgpatSKO* than in control mice. Also, c-Fos immunolabelling and electrophysiological recordings both showed that hypoglycemia activated a lower number of AgRP neurons in *AgRP^AgpatSKO* than in control mice. Thus, *Agpat5* in AgRP neurons plays a role in hypoglycemia sensing but not in whole body insulin sensitivity. The effect of inactivating *Agpat5* on glucagon secretion was seen in both male and female mice. Insulin-induced glucagon secretion shows a sexual dimorphism[23,24] with a stronger response in female mice, but *Agpat5* inactivation does not seem to affect differentially the glucagon secretion between sexes.

To determine the mechanisms by which *Agpat5* could modify glucose sensing by AgRP neurons, we first investigated the possibility that it regulates mitochondrial dynamics. Indeed, Agpat5 is present at the surface of mitochondria and its enzymatic activity leads to the production of PA, an inhibitor of Drp-1, the main regulator of mitochondrial fission[16,17]. Several reports have shown that mitochondrial dynamics has an important role in controlling AgRP and POMC neurons function and physiological role[18,19,25,26]. For instance, inactivating *Drp1* in AgRP neurons reduces body weight gain as a result of lower food intake and increases energy expenditure[19]. Similarly, inactivation of *Mfn1* or *Mfn2*, which control mitochondria fusion, reduces the electrical activity of AgRP neurons of mice fed a high fat diet and leads to lower body weight gain and adiposity[18]. However, electron microscopy analysis of mitochondrial shape and surface area in AgRP neurons showed no impact of *Agpat5* inactivation, indicating that this gene is unlikely to control mitochondrial dynamics.

We, thus, tested the possibility that *Agpat5* inactivation would lead to a change in FAO. Agpat5 catalyzes the production of PA from lyso-PA and fatty acyl-CoAs; lyso-PA being itself produced from glycerol-phosphate and fatty acyl-CoAs, a reaction catalyzed by a family of glycerol-phosphate-acyltransferases (Gpat1-4)[27]. Previous studies in mouse liver have shown that inactivation of the mitochondria-associated *Gpat1* partitions fatty acyl-CoAs towards

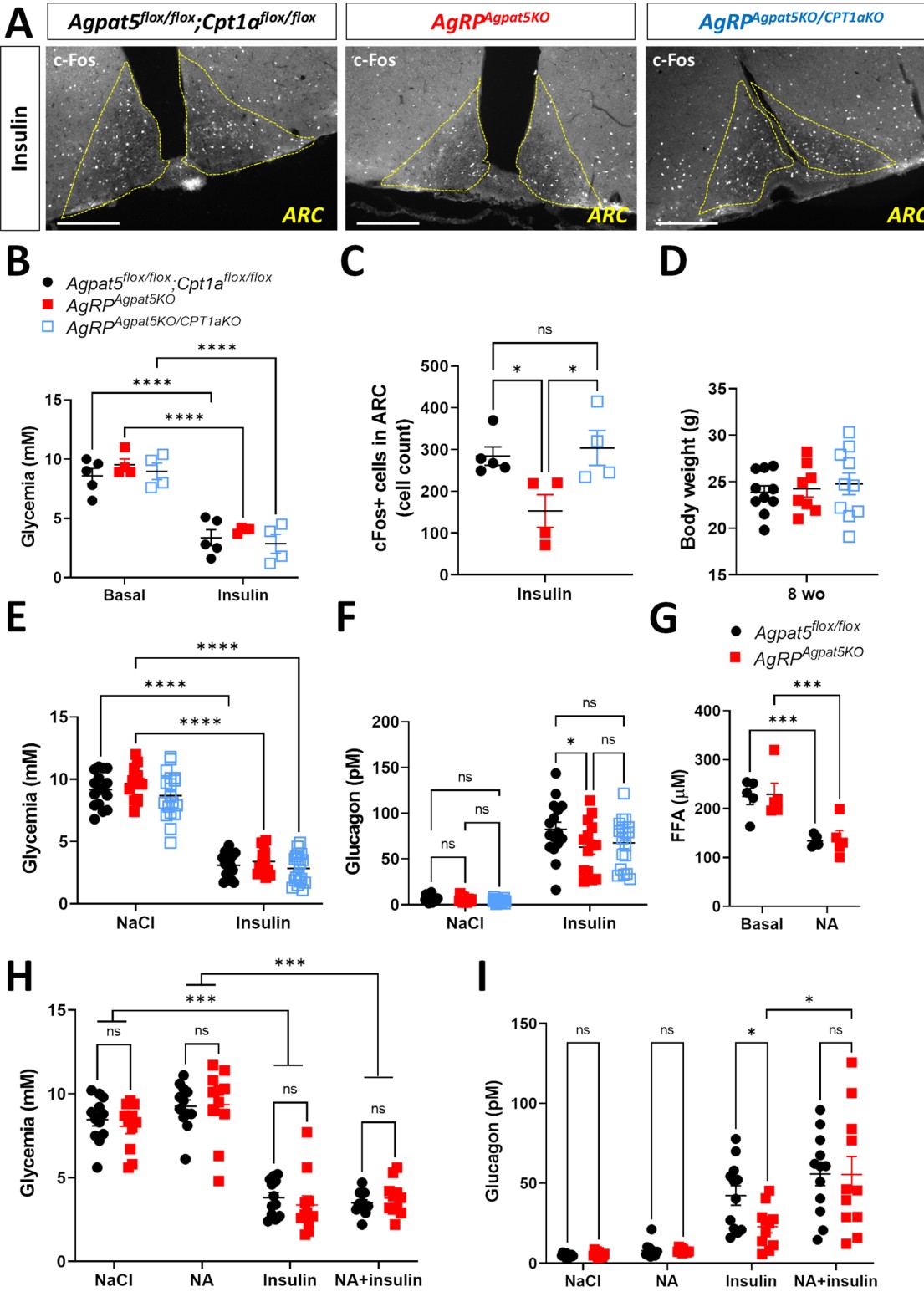

triglyceride synthesis and away from FAO[28,29] but had no effect on mitochondrial morphology[28,30]. Here, we found that *Agpat5* silencing in GT1-7 cells increased OCR and ATP production, an effect that could be blocked by pharmacological inhibition of Cpt1a, or by silencing *Cpt1a*. In mice, inactivation of *Cpt1a* in AgRP neurons of *AgRP^{Agpat5KO}* mice restored their normal activation by hypoglycemia as determined by c-Fos immunostaining and patch clamp analysis. Thus, our data show that the role of Agpat5 is to limit the entry of fatty acids into mitochondria and prevent the FAO-dependent production of ATP. This role is important as AgRP neurons firing is triggered by a fall in intracellular

ATP, which reduces the activity of the Na^+/K^+ATPase leading to membrane depolarization[10,11]. It is therefore essential for these neurons to respond to hypoglycemia that circulating FFAs, which increase during fasting, do not prevent this response by inducing more ATP production. That circulating FFAs indeed blunt glucagon secretion through an *Agpat5*-dependent mechanism was confirmed by showing that the impaired glucagon response to hypoglycemia of *AgRP^{Agpat5KO}* mice could be largely reverted when circulating free fatty acids were lowered by nicotinic acid treatment. A role for *Agpat5* in adaptation to fasting is also congruent with the observation that its expression in

**Fig. 6 | Inactivation of *Cpt1a* or reducing circulating free fatty acids in *AgRP*$^{AgpatSKO}$ mice restores hypoglycemia-activated AgRP neurons and glucagon secretion.** *Agpat5*$^{flox/flox}$;*Cpt1a*$^{flox/flox}$ mice, *AgRP*$^{Agpat5KO}$ mice and *AgRP*$^{Agpat5KO/Cpt1aKO}$ mice were injected with insulin and c-Fos expression in the ARC nucleus was quantitated 1 h later. **A** c-Fos staining in the ARC of the three mouse lines. Scale bar, 200 µm. **B** Glycemic levels. **C** Quantification of c-Fos staining in the ARC of the three mouse lines, *n* = 4–5 mice, three different bregmas/animal. Mice from the three genotypes were injected with NaCl or insulin and plasma glucagon levels were measured one hour later. **D** Body weight. **E** Glycemia 1 h after i.p. NaCl or insulin injection. **F** Plasma glucagon levels. **B**–**F** Data are mean ± SEM, two independent cohorts, *n* = 10–15 mice. \**p* < 0.05, \*\**p* < 0.01 and \*\*\**p* < 0.001, one-way (**C**) or two-

way ANOVA with Tukey's multiple comparisons correction (**B**, **D**–**F**). 5 h-fasted *Agpat5*$^{flox/flox}$ mice and *AgRP*$^{Agpat5KO}$ mice received an i.p. injection of NaCl or nicotinic acid (NA) to reduce plasma free fatty acids levels. 30 min later they received an i.p. insulin injection and plasma glucagon was measured one hour later. **G** Plasma free fatty acid levels before and 30 min after i.p. NA injection, *n* = 5 mice. **H** Glycemia 30 min after NaCl or NA injection and 1 h following insulin injection in NaCl or NA injected mice. **I** Plasma glucagon levels in the conditions described in (**H**). NA-induced decrease in free fatty acids restored glucagon levels in *AgRP*$^{Agpat5KO}$ mice to those of *Agpat5*$^{flox/flox}$ mice, two independent cohorts, *n* = 10–11 mice. **G**–**I** Data are mean ± SEM. \**p* < 0.05, \*\**p* < 0.01 and \*\*\**p* < 0.001, two-way ANOVA with Tukey's multiple comparisons correction.

AgRP neurons is increased two-fold by fasting[31]. It is interesting to note that in the experiment aimed at testing the effect of inactivating *Cpt1a* in *AgRP*$^{Agpat5KO}$ mice the glucagon response to hypoglycemia was only mildly reduced in the *AgRP*$^{Agpat5KO}$ mice (Fig. 6). This could be explained by the fact that the *Cpt1a*$^{lox/lox}$ mice came from a different facility and were on a C57BL6/J and not C57BL6/N background. As all the mice used for this experiment were littermates their genetic backgrounds may have been slightly different from the Control and *AgRP*$^{Agpat5KO}$ mice used in the rest of the study. This indicates that the role of *Agpat5* in AgRP neurons on glucagon secretion depends on epistatic interactions, in line with the results of the genetic screen performed in BXD mice.

As inactivating *Agpat5* in AgRP neurons impairs hypoglycemia sensing and reduces glucagon secretion, this indicates that the AgRP GI neurons act as first order neurons. This can be contrasted to the GI neurons of the VMN. Indeed, inactivation of both α1 and α2 subunits of AMPK in VMN neurons suppresses GI neuron activity in this nucleus but does not impair hypoglycemia-induced glucagon secretion[8]. This can be explained by the fact that VMN GI neurons are part of a neuronal circuit that contains afferent hypoglycemia sensing neurons located in peripheral locations such as the portal vein or the parabrachial nucleus[32,33]. Thus, as long as first order glucose sensing neurons function normally in this circuit, the VMN neurons glucose sensing properties are dispensable for the normal counterregulatory response.

An intriguing observation is that inactivation of *Agpat5* reduces the number of AgRP neurons activated by hypoglycemia without changing the electrophysiological properties of the remaining GI neurons and without reducing the total number of AgRP neurons. This suggests that AgRP GI neurons consist of different subpopulations, which differ in their requirement for *Agpat5* expression for hypoglycemia sensing. These different subpopulations may project to different second order neurons as the *Agpat5*-dependent GI neurons are required for the normal glucagon response but not the *Agpat5*-independent ones. It is known that separate AgRP neuron populations project to different brain regions and have different regulatory functions. For instance, only those AgRP neurons that project to the PVN, lateral hypothalamus, bed nucleus of the stria terminalis, and para-ventricular thalamus (PVT) stimulate food intake whereas those projecting to the central amygdala, and periaqueductal gray do not[34]. In addition, *AgRP*$^{Agpat5KO}$ mice had normal feeding patterns as measured over a 24 h *ad libitum* feeding period or over a 4 h refeeding period following a 16 h fast. Thus, the *Agpat5*-dependent GI neurons may project to nuclei that control autonomic nervous activity, such as the PVN or bed nucleus of the stria terminalis[4,35] and may not be involved in feeding control. Uncovering the molecular basis for the functional diversity of AgRP neurons and identifying their projections to pre-autonomic regions will require further investigations.

In summary, our study identified *Agpat5* expressed in AgRP neurons as required for hypoglycemia-sensing by a population of AgRP neurons and for efficient triggering of glucagon secretion. *Agpat5* role is to divert fatty-acyl CoA away from *Cpt1a*-dependent entry into mitochondria to prevent FAO and ATP generation. This is necessary for intracellular ATP levels to reflect the fall in

extracellular glucose concentrations since AgRP neuron firing is triggered by a decrease in Na$^+$/K$^+$-ATPase activity. Thus, *Agpat5* serves a protective function that ensures that AgRP neurons firing reflects extracellular glucose concentrations, independently of changes in circulating free fatty acids. This is especially important during the fasting period when glucagon secretion needs to prevent hypoglycemia development but when circulating free fatty acids increase. Finally, the results of this and our previous genetic studies[5,36–38] have led to the identification of several genes, acting in different hypothalamic and brainstem neuronal populations, which coordinate the CRR. This enhanced description of the complexity of hypothalamic hypoglycemia sensing will pave the way for new studies aimed at understanding the molecular basis of the impaired CRR of insulin-treated diabetic patients.

## Methods

### Ethical compliance statement
All procedures were conducted in accordance to the Swiss National Institutional Guidelines of Animal Experimentation (OExA; 455.163) with licenses approval (VD3363 and VD3686) issued by the Veterinary Office of Canton de Vaud (Vaud, Switzerland).

### Mice
Experiments were performed with 5- to 20-weeks old male and female mice. *Agpat5*$^{flox/flox}$ mice were generated by genOway (genOway, Lyon, France) by homologous recombination in C57BL6/N embryonic stem cells. A conditional knockout was created by flanking exon 3 of *Agpat5 with loxP* sites, allowing for Cre-dependent recombination. Exon 3 contains a catalytic motif I sequence of Agpat5 protein (HXXXXD) and its deletion results in out of frame splice and introduction of premature STOP codon in exon 4. The resulting truncated mRNA product is unstable and no functional Agpat5 protein is produced. Chimeric males were crossed with C57BL6/N females to establish *Agpat5*$^{flox/flox}$ mouse line. *Sim1-Cre*$^{tg/+}$ mice were a kind gift from Dr. Daniela Cota (University of Bordeaux, France). *CPT1a*$^{flox/flox}$ mice were generated by Prof. Peter Carmeliet (Katholieke Universiteit Leuven, Belgium) and kindly donated by Dr. Marlen Knobloch (University of Lausanne, Switzerland). *Agpat5*$^{flox/flox}$ mice were crossed with *AgRP-Cre*$^{tg/+}$ or *Sim1-Cre*$^{tg/+}$ mice to generate *AgRP*$^{Agpat5KO}$ or *Sim1*$^{Agpat5KO}$ mice and control littermates. For the generation of double *Agpat5* and *Cpt1a* knockout mice (*AgRP*$^{Agpat5KO;CPT1aKO}$), *AgRP-Cre*$^{tg/+}$;*Agpat5*$^{flox/flox}$;*Cpt1a*$^{flox/+}$ male mice were bred with *AgRP-Cre*$^{+/+}$;*Agpat5*$^{flox/flox}$;*Cpt1a*$^{flox/+}$ females. For electro-physiological and electron microscopy studies mice were also crossed with *Rosa26-tdTomato* mice in order to identify AgRP-positive neurons (*AgRP*$^{Cre/+}$;*Rosa26*$^{tdTom}$ and *AgRP*$^{Agpat5KO}$;*Rosa26*$^{tdTom}$ littermates). All mice were back-crossed on a C57BL/6N background for at least 6 generations. All studies used age-matched littermates, randomly assigned to experimental groups. Animals were housed on a 12 h light/dark cycle, at a temperature range of 22–24 °C with the relative humidity kept at 45–55%, with *ad libitum* access to water and food (standard chow SP-150; SAFE, Germany). Mice animal welfare was regularly monitored. For euthanasia, mice were first anaesthetized with isoflurane before by cervical dislocation.

## Mouse genotyping

Mouse genotyping was performed by PCR analysis (*Agpat5* wild-type allele forward- CTACTTTGCTCAGGTAAACTTTGTCTTTGCCC, reverse- CTGAAAATTGTCCTTCGACCTCACATGC; *Agpat5* lox knock-in allele forward- CTGAAAATTGTCCTTCGACCTCACATGC, reverse-TCCTAC ATAGTTGGCAGTGTTTGGGGC; *AgRP-Cre* forward-CCGCAGGTGTAGA GAAGGCA, reverse-CATCCTTAGCGCCGTAAATCAATC; *Agpat5 3exon excision* forward-GGTCTCCAACTCCTAATCTCAGGTGATCTACC, reverse-CTGCTACTGGACTTAAATATACGGACAACCAGC sequence primers. Analysis of genetic recombination in brain regions was performed with DNA extracted (DNeasy Kit, cat: 69504; Qiagen) from micro-dissected 1 mm-thick tissue sections.

## eQTL mapping

eQTL mapping for *Agpat5* was performed using GeneNetwork (mm9 genome) using hypothalamic RNASeq data obtained from the 36 BXD mouse lines used for QTLs mapping of the glucagon secretion trait[5]. The RNASeq data from subcutaneous white fat and liver from BXD mice have been previously published[39,40].

## Immunofluorescence microscopy and in situ hybridization

Mice were deeply anesthetized with isoflurane and thensubjected to a trans-cardiac perfusion of 0.9% NaCl solution followed by 4% cold paraformaldehyde (PFA) in sodium phosphate (PB) buffer (0.1 M, pH 7.4). Brains were isolated and kept for 3 h in 4% PFA at 4 °C followed by overnight incubation in sucrose 20% at 4 °C and freezing in dry-ice. Samples were stored at −80 °C until 25 μm cryo-sections were prepared and placed on glass slides. Slides were washed three times with PBS and incubated in a blocking buffer solution (4% (v/v) of normal goat serum (NGS) and 0.3% Triton X-100 in PBS) for 1 h. Then, slides were incubated with a monoclonal rabbit anti-cFos antibody (#2250, 1/7000, Cell Signaling, Danvers, USA) diluted 1:1000 in blocking solution for 16 h at room-temperature, washed three times with PBS and incubated in AlexaFluor568-conjugated goat anti-rabbit IgG antibodies (#A-11011, Invitrogen) diluted 1:500 in blocking solution for 3 h. Nuclei were stained using DAPI nuclear stain (D1306, Invitrogen) diluted 1:10,000 in PBS for 10 min in the dark. Slides were mounted in Mowiol for fluorescence imaging and imaged using Zeiss Axio Imager M2 microscope, equipped with ApoTome.2 and a Camera Axiocam 702 mono (Zeiss, Oberkochen, Germany). For each mouse, images of the ARC at 3 bregmas were taken and c-Fos-positive cells were counted using ImageJ.

For stereological analysis of AgRP/tdTomato-positive cell numbers in *AgRP^{Cre/+};Rosa26^{tdTom}* and *AgRP^{Agpat5KO};Rosa26^{tdTom}*, three 70 μm-thick ARC sections at three different bregmas were used. Slides were imaged using z-stack with 40× objective and cell number was quantified along the z-stack using ImageJ.

For in situ hybridization analysis, the hypothalamus was dissected and fixed for 28 h in 10% formalin before embedding in paraffin. Five μm sections were prepared and in situ hybridization for *Agpat5* mRNA (#514011, bio-techne, UK) was processed using the RNAscope and RNAscope 2.0 HD Red Detection Kit Assay (Advanced Cell Diagnostics) and counterstained with Mayer's hematoxylin. Sections were mounted using Aquatex mounting medium (# 363123S, VWR), and imaged using an Axio Imager D1 (Zeiss) microscope interfaced with AxioVision software (Zeiss).

## Transmission electron microscopy analysis

Male *AgRP^{Cre/+};Rosa26^{tdTom}* and *AgRP^{Agpat5KO};Rosa26^{tdTom}* (12 weeks old, *ad libitum* fed) were subjected to a trans-cardiac perfusion of fresh cold fixative solution (1% glutaraldehyde/2% cold PFA in 0.1 M phosphate buffer, pH 7.4). Brains were isolated and kept overnight in the fixative solution. After two washes with PB solution, 70 μm coronal sections containing ARC were prepared using a vibratome (VT1000S, Leica) and tdTomato-positive fluorescent cells were imaged using a ZEISS Axio Imager.M2 microscope, equipped with ApoTome.2 and a Camera

Axiocam 702 mono using the AxioVision software (Zeiss, Oberkochen, Germany). Sections were osmicated (1.5% osmium tetroxide) for 1 h followed by 1 h incubation in 1% (wt/vol) tannic acid in 0.1 M PB buffer. Samples were then dehydrated with ethanol solutions (1% uranyl acetate in 70% ethanol for 30 min) and flat-embedded in Epon-araldite. Trimmed blocks were used to cut ultrathin sections that were examined using a Philips CM-10 electron microscope[41]. Electron microscopy images were cross-referenced with immunofluorescent images of tdTomato-positive AgRP neurons to identify the cell population of interest. Mitochondrial morphology in AgRP neurons was analyzed using ImageJ software, quantifying mitochondrial area and perimeter in tdTomato-positive AgRP neurons[42].

## Stereotaxic surgery and fiber photometry

Stereotaxic surgeries were performed under ketamine/xylazine anaesthesia. 8 week-old male mice were placed on a stereotaxic frame (Stoelting, Wood Dale, IL, USA) and the surface of the skulls was visualized with a digital microscope (DMS 300, Leica Microsystems GmbH, Wetzlar, Germany). Recombinant adeno associated virus (AAV) were infused unilaterally in ARC (coordinates relative to Bregma: −1.7 mm anteroposterior, −0.30 mm mediolateral, and −5.8 mm dorsoventral) at a rate of 100 nl/min using a 33-gauge stainless steel injector (Hamilton Company, Reno, NV, USA) and a microsyringe pump controller (Micro4, World Precision Instruments, Sarasota, FL, USA). AAV9-syn-FLEX-jGCamp7s-WPRE was purchased from Addgene (Cat#104491-AAV9). Optical fiber cannulas (CFML12U, Thorlabs Inc. Newton, NJ, USA) were positioned unilaterally in ARC (coordinates relative to Bregma: −1.7 mm anteroposterior, −0.30 mm mediolateral, and −5.8 mm dorsoventral) and fixed to the skull with surgical screws (P1 Technologies, Roanoke, VA, USA), tissue adhesive (VetBondTM; 3MTM, Saint Paul, MN, USA) and dental cement (Paladur; Kulzer GmbH, Hanau, Germany). Mice were placed in individual cages on heating pads (at 37 °C) for 1–2 h after surgery in order to prevent hypothermia. Animals were allowed to recover for 2 weeks in individual cages with daily handling and body weight monitoring.

The inhibition of GCaMP7 fluorescence upon refeeding was used as a positive control for the responsiveness of AgRP neurons and validation of cannula implantation. For this, naïve mice were fasted for 16 h (overnight) and presented with Chow pellets, and GCaMP7 signal reduction was assessed. For i.p. 0.9% NaCl and insulin (0.8 U/kg) injections, mice were fasted for 5 h, GCaMP7 baseline activity was recorded for 30 min before and for 1 h after the injection. Glycemia was measured at the beginning of recording and 1 h after the injection. Animals were allowed to recover for 5–7 days between recordings.

Fiber photometric recordings were performed using optical components from Doric lenses controlled by Tucker Davis Technologies fiber photometry processor RZ5P, with Doric software used for data acquisition. Emission in 405 nm channel was used as an isosbestic control and GCaMP7-specific emission was recorded in 465 nm channel. The standard fiber photometry analysis pipeline was automated using a Python code and *Jupyter* interface. Briefly, raw traces were uploaded and the isosbestic channel was subtracted from the GCaMP7 channel, photo-bleaching was removed by subtracting an exponential fit curve, and data was presented as $\Delta F/F$ and Z-score. Recordings were normalized to 10 min baseline before the i.p. NaCl/insulin injection, butterworth low-pass filter at 1 Hz was applied to remove high-frequency noise and 500-step decimation was applied to the original signal (recorded at 123 Hz) to facilitate statistical analysis. The code is available at https://gitlab.com/pcanilho/fiber-photometry-analysis.

## Electrophysiology

Glucose responsiveness of ARC AgRP neurons was assessed by electrophysiology. Five to 7 weeks old *ad libitum* fed male mice were deeply anesthetized with isoflurane before decapitation, and 250 μm coronal sections containing ARC were sliced using a vibratome

(VT1000S, Leica). Electrophysiological recordings were conducted in slices were placed in a submerged-type recording chamber and continuously superfused with extracellular solution at room temperature[38]. tdTomato-positive neurons were identified by fluorescence microscopy. Whole-cell recordings were performed in current-clamp mode using a MultiClamp 700B amplifier coupled to a 1440A Digidata digitizer (Molecular Devices). Neurons exhibiting an access resistance over 25 MΩ or changed by more than 20% during the recording were discarded from analysis. A hyperpolarization step (−20 pA, 500 ms) was applied every 30 s to assess membrane resistance. Membrane potential and resistance were monitored over time under 5 mM (normoglycemia) or 0.1 mM (hypoglycemia) extracellular glucose concentration after a 10–15 min baseline. Signals were digitized at 10 kHz, collected and analyzed using the pClamp 10 data acquisition system (Molecular Devices). Membrane potential and resistance values during the last 2.5–3 min (5–6 data points) of the low glucose application (0.1 mM) were compared to 5–6 data points monitored during baseline condition (5 mM glucose). Using a two-tailed paired t-test, a neuron was considered as GI if it exhibited a significant increase of its membrane potential (depolarization) after switching to 0.1 mM glucose and if it recovered its initial membrane potential when glucose was switched back to 5 mM. Significance was assessed using a paired t-test.

## Autonomous nervous system activity recording
Unipolar parasympathetic activity was recorded in 5 h-fasted 10–13 weeks old mice, measuring the firing rate activities of the thoracic branch of the vagus nerve along carotid artery[43]. Recordings were performed for one and half hour under isoflurane anesthesia (for 30 min during basal condition then for 1 h following i.p. insulin (0.8 U/kg) injection) using the LabChart 8 software (AD Instrument, Oxford, UK). Data were digitized with PowerLab 16/35 (AD Instrument, Oxford, UK). Signals were amplified 105 times and filtered using 100/1000 Hz band pass filter. Firing rate analysis was performed using LabChart 8.

## Glucose and insulin tolerance tests
Glucose tolerance test was performed in 8 weeks old male and female mice injected with i.p. glucose (2 g/kg) after 16 h fasting. Insulin tolerance was assessed in 12–13 weeks old male and female mice injected with i.p. insulin (0.8U/kg) after 5 h fasting, and glycemia was measured from tail blood taken at the indicated times after injection[44].

## Glucagon secretion and hyperinsulinemic-hypoglycemic clamp
Glucagon secretion was assessed in 10 weeks old male and female mice subjected to insulin-induced hypolgycemia (IIH). Mice were fasted for 6 h and blood was collected 1 h after i.p. saline or insulin (0.8 U/kg) injection into micro-centrifuge tubes containing aprotinin-A/EDTA. Inadequate i.p. injection or health problem were used as exclusion criteria. Plasma glucagon concentrations were assessed by ELISA (Mercodia, Uppsala, Sweden). Nicotinic acid (NA, 32 mg/kg, i.p.) injection was performed in 5 h-fasted males between 11 and 15 weeks of age. Free fatty acids (FFA) concentration in plasma was assessed using FFA Assay kit (abcam, #ab65341). Hyperinsulinemic-hypoglycemic clamps were performed in 15–16 weeks-old mice, with a 33% glucose solution being infused for 150 min through either the portal or femoral vein at an insulin injection rate of 18 mU/kg/min[45], after 5 h of fasting.

## Cell culture
GT1-7 cells were a kind donation from Dr. Pamela L. Mellon's laboratory. GT1-7 cells were cultured (37 °C, 5% $CO_2$) in DMEMGlutaMax (25 mM glucose, 1 mM sodium pyruvate, 2 mM L-glutamate) supplemented with FBS 10% (v/v) and horse serum 5% (v/v). HEK293T cells were purchased from ATCC (Manassas, USA). For mitochondrial function experiments, confluent GT1-7 cells were cultured for 16 h in 0.1 mM glucose, serum-free DMEM supplemented with 25 mM mannitol to control for osmolality.

## Cells transfection
GT1-7 cells (~70% confluence) or HEK293T cells (~80% confluence) in six-well plates were incubated in 0.9 ml/well Opti-MEM dishes. Plasmid DNA−Lipofectamine 3000 complexes (150 μl of 2 μg plasmid DNA, 2.5% (v/v) Lipofectamine 3000, 1% P300 (v/v)) for HEK293T cells and/or siRNA-Lipofectamine RNAiMAX (150 μl of 200 nM siRNA, 2.5% (v/v) Lipofectamine RNAiMAX) for HEK293T and GT1-7 cells were added dropwise to each well. Cells were incubated at 37 °C for 3 h before the transfection media were supplemented with complete culture media and cultured for 48 h.

## Preparation of cell lysates
Cells were washed with phosphate buffered saline (PBS, Gibco, 10010023) prior to the addition of ice-cold lysis buffer (50 mM Tris-HCl, pH 7.4 at 4 °C, 150 mM NaCl, 50 mM NaF, 1 mM $Na_4P_2O_7$, 1 mM EDTA, 1 mM EGTA, 1% (v/v) Triton-X-100, 250 mM mannitol, 1 mM DTT, 1 mM $Na_3VO_4$, 0.1 mM benzamidine, 0.1 mM phenylmethylsulphonyl fluoride). Cell lysates were scraped into microcentrifuge tubes, incubated on ice for 20 min, centrifuged (5 min, 21,910 × g, 4 °C) and the subsequent supernatants stored at −20 °C. Lysate protein concentrations were determined using bicinchoninic acid (BCA) method.

## Immunoprecipitation
Agpat5 and GFP were immunoprecipitated from 0.5 mg of cell lysate using 2 μl of anti-Agpat5 or anti-GFP antiserum, or pre-immunization rabbit serum in 500 μl IP (immunoprecipitation) buffer (50 mM Tris-HCl, pH 7.4 at 4 °C, 150 mM NaCl, 50 mM NaF, 1 mM $Na_4P_2O_7$, 1 mM EDTA, 1 mM EGTA, 0.1% (v/v) Triton-X-100, 1% (v/v) glycerol, 1 mM DTT, 1 mM $Na_3VO_4$, 0.1 mM benzamidine, 0.1 mM phenylmethylsulphonyl fluoride) in a 1.5 ml microcentrifuge tube. After overnight agitation at 4 °C, Protein A agarose beads were added and incubated for 1 h with agitation. Samples were centrifuged for 30 s in a bench top centrifuge and the supernatants were collected. Beads were washed four times with 1 ml IP buffer. After the final wash, excess IP buffer was removed and 2 μl of 4X SDS-PAGE sample buffer added to each sample.

## Western blot analysis
Proteins were resolved by SDS-PAGE, transferred to nitrocellulose membranes and immunodetected with antibodies diluted in TBS (Tris-Buffered saline, 20 mM Tris-HCl, 150 mM NaCl, pH 7.4) containing 0.1% (v/v) Tween-20 and 5% (w/v) (TBSt) BSA[46]. Briefly, nitrocellulose membranes were incubated for 16 h at 4 °C with a custom-generated, affinity-purified rabbit anti-Agpat5 antibody (1/1000 in 3% BSA/PBS (w/v), BioTem, France) and a mouse anti-α-tubulin (1/5000 in 3% BSA/PBS (w/v), T5168, Sigma-Aldrich). Then, membranes were washed and incubated with secondary antibodies for 1 h at room temperature, using infrared dye-labeled (donkey anti-rabbit #926-32213 or donkey anti-mouse #926-48072, LI-COR, Cambridge, UK) secondary antibodies (in 3% BSA, TBSt, for 1 h). Proteins were then revealed using a FusionXF (Vilber, France) infrared imaging system and densitometric analysis of band intensity was performed with *ImageJ*.

## Mitochondrial respiration and assessment mitochondrial of fatty acid β-oxidation
Oxygen consumption rate (OCR) in GT1-7 cells was measured using a Seahorse XFe24 or a Seahorse XFe96 instrument. GT1-7 cells were seeded into 96-well plates at a density of $5 \times 10^4$ cells/well or 24-well plates at a density of $2 \times 10^5$ cells/well. When 70% confluent, cells were transfected with siRNAs, as described above. After 48 h, the medium was changed to 0.1 mM glucose DMEM and then incubated in phenol-free DMEM medium (0.1 mM glucose, 1 mM pyruvate, 1.5 mM L-carnitine, 2 mM L-glutamine, pH 7.4, HEPES-free), with vehicle or etomoxir (50 μM) for 1 h at 37 °C prior to the assay. Basal respiration was recorded for 30 min, and following the sequential addition of glucose (5 mM),

oligomycin (1 μM), FCCP (1 μM), rotenone (0.5 μM) and antimycin A (1 μM) to assess maximal and non-mitochondrial respiration[47]. Oxygen consumption was normalized to total cellular protein levels. Analysis was performed using Wave Softwarev2.4.2 (Seahorse, Agilent).

### RNA extraction and gene expression analysis
RNA was extracted from GT1-7 cells or micro-dissected brain tissues using a RNeasy kit (Qiagen, Hombrechtikon, Switzerland) and reverse-transcribed (1 μg RNA) using a High Capacity cDNA Reverse Transcription kit (Applied Biosystems). qPCR was performed with an Applied Biosystems ABI-PRISM 7500Fast Sequence Detection System. *Agpat5* gene expression was normalized to *β-actin* using PowerUp Sybr green master mix (Applied Biosystems) and custom oligonucleotide primers (*Agpat5* forward- GCTTCCGCTGTATGGGTTCT, reverse-CATCGGT-GTTCCTGCGTTCA; *Cpt1a* forward- ACGTTGGACGAATCGGAACA, reverse-GGTGGCCATGACATACTCCC; and *β-actin* forward-CTAAGG CCAACCGTGAAAAGAT, reverse-CACAGCCTGGATGGCTACGT). Relative levels of mRNA were assessed using the *ΔΔCt* method.

### Statistical analysis
All data is expressed as mean ± SEM. Prism software 9.0.1 (GraphPad Software, San Diego, CA) was used to perform student t-tests, one or two-way ANOVA followed by an appropriate post hoc test, where appropriate, with $p < 0.05$ considered significant. For comparison between proportions of different groups of glucose-sensing neurons in electrophysiological experiments Fisher's exact test was used, with $p < 0.05$ considered significant.

### Reporting summary
Further information on research design is available in the Nature Research Reporting Summary linked to this article.

## Data availability
The raw data (Fastq files) for hypothalamus RNAseq data used in this study are available in the Gene Expression Omnibus (GEO) database under accession code GSE87586. The data sets for liver[40] and white adipose tissue[39] RNAseq used in this study are available in the Gene Expression Omnibus (GEO) database under accession code GEO: GSE114845 and GEO: GSE79016, respectively. The mouse metabolic phenotyping and electrophysiology data generated in this study are provided in the Source Data file. The raw data for each individual experiment are available upon request to the corresponding author. Source data are provided with this paper.

## Code availability
The code for fiber photometric data analysis is available at https://gitlab.com/pcanilho/fiber-photometry-analysis.

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

## Acknowledgements

We thank Prof. Paul Franken (UNIL, Lausanne) for liver RNAseq data. We are grateful to the Mouse Metabolic Analysis facility (MEF) of CIG, University of Lausanne for hyperinsulinemic-hypoglycemic clamp experiments and to the Electron Microscopy Facility (EMF) of Uni-versity of Lausanne for transmission electron microscopy sample processing and imaging. We thank Prof. Peter Carmeliet (Katholieke Universiteit Leuven, Belgium), and Prof. Marlen Knobloch (University of Lausanne) for *Cpt1a^{flox/flox}* mice. We are grateful to Dr. Sevasti Gas-pari, Dr. Sophie Croizier, Dr. Simon Quenneville, Dr. Isabel C. Lopez-Mejia, Paulo Canilho and Manon Gervais for useful discussions and technical guidance for this study. We are also grateful to Dr. Quen-neville for cloning of murine *Agpat5* cDNA. This work was supported by a European Research Council Advanced Grant (INTEGRATE, No. 694798) and a Swiss National Science Foundation grant (310030-182496) to B.T., and has received funding from the Innovative Medi-cines Initiative 2 Joint Undertaking (JU) under grant agreement No 777460 (HypoRESOLVE). The JU receives support from the European Union's Horizon 2020 research and innovation program and EFPIA and T1D Exchange, JDRF, International Diabetes Federation (IDF), The Leona M. and Harry B. Helmsley Charitable Trust.

## Author contributions

B.T. conceived the project. B.T. and A.S. designed and performed the experiments and interpreted the data. A.S., G.L., A.P., X.P.B., and D.T. performed experiments and analyzed data. M.J. performed bioinfor-matic analysis. B.T. and A.S. co-wrote and edited the manuscript.

## Competing interests

The authors declare no competing interests.
