## [Peer Review File · Nature Communications]

Lipid biosynthesis enzyme Agpat5 in AgRP-neurons is required for insulin-induced hypoglycemia sensing and glucagon secretionREVIEWER COMMENTS

Reviewer #1 (Remarks to the Author):

This manuscript describes the results from experiments designed to determine the role that *Agpat5* plays in hypoglycemia-induced glucagon secretion. This follows the identification of this gene as a candidate regulator previously by this group. Using a variety of transgenic mouse models they demonstrate that *Agpat5* expressed in hypothalamic (arcuate) AgRP neurons (not VMH or LH) is required for AgRP neuron depolarisation and activation by hypoglycemia and for glucagon secretion. In addition, using a hypothalamic cell line they have shown that *Agpat5* negatively regulates fatty acid oxidation allowing a decrease in ATP levels during a hypoglycemic challenge. Finally, they showed in mice that preventing or reducing fatty acid oxidation in AgRP neurons restored AgRP neuron responsiveness to hypoglycemia and recovered glucagon secretion in AgRP*Agpat5* KO mice.

I have a few comments that the authors may wish to address:

1. In Figure 2C and D, the response to insulin appears to be more affected in males vs females with respect to the AgRP*Agpat5* KO mice. As the clamps shown were performed in males only (or are there data for female mice also?) it is difficult to assess whether there is sexual dimorphism associated with the glucagon response in these mice. Can you please comment on this possibility in the Discussion?
2. Figure 2F and G indicates possible increased insulin sensitivity from the clamp studies in the male AgRP*Agpat5* KO mice. Comment?
3. Figure 4 D and F – fibre photometry. Figure 4F shows an increase with time for the AgRP*Agpat5* KO mice after insulin injection but this is only compared with the same time point for the control mice. Is this increase significant compared to the previous time segment (50-70minutes) for these AgRP*Agpat5* KO mice? Please also clarify that these are representative traces/recordings (or is this the mean of all traces?) in the legend to figure.
4. Figure 5F. This is quite a busy graph with 6 datasets presented – could make this into two graphs ± etomoxir? Also please clarify in text/legend this is a mito stress test and also please add the nutrients (and concentrations) that the cells were exposed to in this experiment (in Methods) – glucose, pyruvate, glutamine?
5. Page 8 OCR – could you add a little explanation for the change in ATP production linking with OCR using oligomycin for the general reader?
6. GT1-7 cells as a model for AgRP neurons? The general view is that GT1-7 cells model GE neurons rather than GI? Comment in Discussion?
7. Was food intake determined in this study – comparing AgRP*Agpat5* KO mice and the floxed controls? The comment in the Discussion regarding fasting increasing *Agpat5* expression, if applied to AgRP neurons would suppose that FAO would be limited. If so, how does ghrelin act on these neurons to increase food intake etc – as this is argued to be a FAO-dependent process – does ghrelin over-ride the *Agpat5* effect? Can you comment on this aspect in the Discussion?
8. In Methods Page 19. GI neuron WCR – neurons depolarise after switching to 0.1 mM glucose, then recovery. Can you add what criteria was used to determine whether any given neuron gave a positive response (depolarisation) – what was the minimum ΔV acceptable from baseline average prior to switch – 3xSD of previous baseline?

Minor points:

1. Figure 2F – GINF 2 way ANOVA used (not repeated measures?) and no asterisks showing

significance on figure.

2. Figure 2G and Figure 3F– state statistical test used.

Reviewer #2 (Remarks to the Author):

The authors previously performed a genetic screen to identify novel hypothalamic regulators of insulin-induced glucagon secretion. They found QTLs on chromosome 8 and 15. *Irak4* was identified in the QTL of chromosome 15. They now wished to investigate a candidate gene of the chromosome 8 QTL, namely *Agpat5* ((1-acylglycerol-3-phosphate-O-acyltransferase 5) in glucagon secretion. Inactivation of *Agpat5* in AgRP neurons of the hypothalamus suppressed their activation by hypoglycemia, impaired hypoglycemia-induced vagal nerve activity, and reduced glucagon secretion. In a hypothalamic cell line and in mice *Agpat5* appeared to is to divert free fatty acids away from Cpt1a-dependent entry in the mitochondria, FAO, and ATP production. Thus a fall in glycemia leads to decreases in intracellular ATP and AgRP neuron firing.

These are interesting and comprehensive studies, and the authors elegantly maneuver between different experimental models to map the details of the involved pathways and mechanisms. The manuscript is clearly written and the data well presented. My only reservation is that the changes in glucagon secretion are rather unimpressive. In addition, one must consider the direct glycemia driven changes in glucagon secretion. So, how much does the hypothalamic mechanism really matter? I also find it surprising that insulin sensitivity and amounts of glucose needed in clamps are all the same in spite of the reported changes in glucagon responses. It is compelling that the *cfos* staining in responses to hypoglycemia depended on *Agpat5* in the AgRP neurons of the ARC as well as the results from patch clamp analysis and Ca⁺ recordings. Vagal activity was found to be increased, but it is not demonstrated that this actually leads to glucagon secretion. One would have expected that also the sympathetic system would be engaged in the response.

The case is made regarding mechanisms that the role of *Agpat5* is to divert fatty-acyl-CoAs from Cpt1-mediated entry into mitochondria and this was supported by inactivation of *cpt1a* which restored normal responsiveness. (but again the glucagon levels were not widely different).

I. 40 GI (neurons) is used as an abbreviation for “glucose inhibited” ; this may be confusing since it often means “gastro-intestinal”

I. 43. In humans the main stress induced inhibitor of insulin secretion is the noradrenergic innervation of the pancreas (not the hormones) and probably also the stimulation of glucagon secretion .

Reviewer #3 (Remarks to the Author):

This paper describes a role of *Agpat5* in AgRP neurons in sensing insulin-induced hypoglycemia and related elevated glucagon secretion. In my view, the overall message is novel and important. I find many of the studies well designed, executed and interpreted. However, there are several results that are incomplete or do not address the critical questions.

1) The paper is about the counter regulatory response of exogenous insulin-induced hypoglycemia. This should be specified in the title. There is hypoglycemia associated with fasting, which is not the focus of the paper. Or if it is the intent of the authors to include all types of shifts in glucose levels in the hypothalamus, they should specify], which experiments is related to which type of hypoglycemia.

2) The CPT1a work is not consistent with the way these neurons function under fasting conditions, which is the reason I suggest that the authors clearly specify that their focus is insulin-triggered hypoglycemia.

3) The analysis of mitochondrial parameters in the AgRP-specific *Agpat5* KO animals is incomplete, and as it stand, the data is not informative. They should have looked at mitochondria in these animals in the various experimental conditions. On the other hand, these studies are not critical for the main focus and could be dropped.

4) References 10 and 11 are not supportive of the declarative statement regarding how AgRP

neurons are activated during hypoglycemia. Similar to the point above, these are not pertinent to the main finding of the paper.

5) Finally, if the paper is about insulin-induced hypoglycemia, which is clearly a crucial medical issue, it would be nice to see how this mechanism is relevant in paradigms when the counter-regulatory response fails.

NCOMMS-22-18552-T

***Agpat5* in AgRP neurons is required for insulin-induced hypoglycemia sensing and glucagon secretion**

REBUTTAL

We would like to thank the reviewers for their positive and helpful comments on our manuscript. Detailed point-by-point answers to their remarks and requests are presented below.

REVIEWER COMMENTS

Reviewer #1 (Remarks to the Author):

This manuscript describes the results from experiments designed to determine the role that *Agpat5* plays in hypoglycemia-induced glucagon secretion. This follows the identification of this gene as a candidate regulator previously by this group. Using a variety of transgenic mouse models they demonstrate that *Agpat5* expressed in hypothalamic (arcuate) AgRP neurons (not VMH or LH) is required for AgRP neuron depolarisation and activation by hypoglycemia and for glucagon secretion. In addition, using a hypothalamic cell line they have shown that *Agpat5* negatively regulates fatty acid oxidation allowing a decrease in ATP levels during a hypoglycemic challenge. Finally, they showed in mice that preventing or reducing fatty acid oxidation in AgRP neurons restored AgRP neuron responsiveness to hypoglycemia and recovered glucagon secretion in AgRP*Agpat5* KO mice.

I have a few comments that the authors may wish to address:

1. In Figure 2C and D, the response to insulin appears to be more affected in males vs females with respect to the AgRP*Agpat5* KO mice. As the clamps shown were performed in males only (or are there data for female mice also?) it is difficult to assess whether there is sexual dimorphism associated with the glucagon response in these mice. Can you please comment on this possibility in the Discussion?

Answer: There is indeed a sexual dimorphism in glucagon secretion with female mice secreting more glucagon than male mice upon similar insulin-induced hypoglycemic levels (Figures 2A,C vs. 2 B,D). This dimorphism has already been reported (see for instance: Steinbusch et al., 2016, PMID: 27422385; Mahmood et al., 2018, PMID: 30954669). However, the impact of *Agpat5* inactivation in AgRP neurons on glucagon secretion is approximately the same in male and female mice (reduction of ~20 pM, Fig 2C,D) and, thus, does not show sexual dimorphism. The hypoglycemic clamps have indeed been performed only with male mice and we cannot comment on sexual dimorphism in this experimental setting. We have added the following sentence at the end of the first paragraph of page 12: “The effect of inactivating *Agpat5* on glucagon secretion was seen in both male and female mice. Insulin-induced glucagon secretion shows a sexual dimorphism^{23,24} with a stronger response in females but *Agpat5* inactivation does not seem to affect differentially the glucagon secretion between sexes.”

2. Figure 2F and G indicates possible increased insulin sensitivity from the clamp studies in the male AgRP Δ Agpat5 KO mice. Comment?

Answer: We thank the reviewer for this remark. Figure 2F indicates that there is a (non-significant) difference in GINF between the two mouse groups during the period leading to the steady state (up to 90 minutes). Once the target hypoglycemia is reached (see Figure 2E) the GINF are identical in both groups of mice, indicating identical insulin sensitivity. The difference in the GINF during the first 90 minutes can be due to multiple factors, including differences in induction of glucagon secretion. Figure 2G, presenting the GINF AUC over the 0-90 minute period, and which showed a near significant ($p = 0.053$) difference between groups has now been removed as it was actually uninformative.

3. Figure 4 D and F – fibre photometry. Figure 4F shows an increase with time for the AgRP Δ Agpat5 KO mice after insulin injection but this is only compared with the same time point for the control mice. Is this increase significant compared to the previous time segment (50-70minutes) for these AgRP Δ Agpat5 KO mice? Please also clarify that these are representative traces/recordings (or is this the mean of all traces?) in the legend to figure.

Answer: In figure 4F, there is no increase GCaMP7 fluorescence in the AgRP neurons of the Agpat5 knockout mice (red trace). The GCaMP7 fluorescence over the 70-80 minutes period in AgRP neurons of control mice is significantly higher than over the 40-50 minutes period of the same neurons. This is now indicated in the quantitation of Figure 4G.

We clarified in the figure's legend that Fig 4D and 4F are showing mean \pm SEM of the fiber photometry traces (n=6). We apologize for the confusion.

4. Figure 5F. This is quite a busy graph with 6 datasets presented – could make this into two graphs \pm etomoxir? Also please clarify in text/legend this is a mito stress test and also please add the nutrients (and concentrations) that the cells were exposed to in this experiment (in Methods) – glucose, pyruvate, glutamine?

Answer: We thank the reviewer for his/her suggestions. We have now separated the vehicle and etomoxir groups into two graphs (Fig 5F and 5G). We clarified in the figure legend that the data came from the same experiment.

We also added additional details in the Methods section on assay medium and nutrient concentrations used in the mitochondrial stress test (page 22, second paragraph).

5. Page 8 OCR – could you add a little explanation for the change in ATP production linking with OCR using oligomycin for the general reader?

Answer: We clarified that the production of ATP was calculated from the difference between basal OCR rate and OCR after blocking ATP synthase with oligomycin. This is specified the legend to figure 5 F,G.

6. GT1-7 cells as a model for AgRP neurons? The general view is that GT1-7 cells model GE neurons rather than GI? Comment in Discussion?

Answer: Indeed, the GT1-7 cell line has been previously described as a model of glucose-excited adult hypothalamic neurons (Beall *et al.* 2012, PMID: 22760787). Although, electrophysiological properties of GT1-7 cells may be different from adult AgRP neurons, these cells express NPY and AgRP, sharing several characteristics with AgRP neurons, which is why they are commonly used as a model of AgRP-like murine neurons. Thus, we chose the GT1-7 cell line as a model to perform mitochondrial energetics study and the results obtained regarding the role of *Agpat5* and *Cpt1a* were confirmed in the subsequent studies with double knockout mice.

7. Was food intake determined in this study – comparing AgRP*Agpat5* KO mice and the floxed controls? The comment in the Discussion regarding fasting increasing *Agpat5* expression, if applied to AgRP neurons would suppose that FAO would be limited. If so, how does ghrelin act on these neurons to increase food intake etc – as this is argued to be a FAO-dependent process – does ghrelin over-ride the *Agpat5* effect? Can you comment on this aspect in the Discussion?

Answer: We did not observe differences in food intake over a 24 h period or during a re-feeding experiment in 16 h fasted male mice (see figure below) indicating that *Agpat5* inactivation in AgRP neurons does not impact feeding nor ghrelin action. AgRP neurons are known to project to multiple secondary neurons located in different brain regions (ref 34) and to control various functions, including feeding behavior. AgRP neurons that control glucagon secretion may, thus, represent a subpopulation of AgRP neurons, as discussed in the Discussion, page 13, last paragraph. We have now added a second sentence on top of page 14: “In addition, *AgRP^{Agpat5KO}* mice had normal feeding patterns as measured over a 24 hours *ad libitum* feeding period or over a 4 hours refeeding period following a 16 hour fast (not shown).” And modified the following sentence: “Thus, the *Agpat5*-dependent GI neurons may project to nuclei that control autonomic nervous activity, such as the PVN or bed nucleus of the stria terminalis^{4,35} and may not be involved in feeding control.”

Figure : *AgRP^{Agpat5KO}* mice do not display changes in food consumption over 24h period and differences in food intake during 4 h of refeeding after 16 h fasting.

(A) Food intake in *Agpat5^{flox/flox}* mice and *AgRP^{Agpat5KO}* mice over 24 h period with *ad libitum* access to food.

(B) Cumulative food consumption over 4h of refeeding after a 16 h fasting period in male mice. n=10-12 mice per genotype. Data are mean±SEM.

8. In Methods Page 19. GI neuron WCR – neurons depolarise after switching to 0.1 mM glucose, then recovery. Can you add what criteria was used to determine whether any given neuron gave a positive response (depolarisation) – what was the minimum deltaV acceptable from baseline average prior to switch – 3xSD of previous baseline?

Answer: We clarified the parameters of acquisition and the criteria used for the depolarization in the Methods section page 19, end of second paragraph: “Membrane potential and resistance values during the last 2.5-3 min (5-6 data points) of the low glucose application (0.1 mM) were compared to 5-6 data points monitored during baseline condition (5 mM glucose). Using a two-tailed paired t-test, a neuron was considered as GI if it exhibited a significant increase of its membrane potential after switching to 0.1 mM glucose and if it recovered its initial membrane potential when glucose was switched back to 5 mM.”

Minor points:

1. Figure 2F – GINF 2 way ANOVA used (not repeated measures?) and no asterisks showing significance on figure.

Answer: the GINF data in Fig 2F were analyzed using two-way ANOVA repeated measures with Sidak’s *post hoc* corrections, but no significant differences were identified.

2. Figure 2G and Figure 3F– state statistical test used.

Answer: The statistical tests used have now been added to the legend of Fig 2F, 2G and 3F.

Reviewer #2 (Remarks to the Author):

The authors previously performed a genetic screen to identify novel hypothalamic regulators of insulin-induced glucagon secretion. They found QTLs on chromosome 8 and 15. *Irak4* was identified in the QTL of chromosome 15. They now wished to investigate a candidate gene of the chromosome 8 QTL, namely *Agpat5* ((1-acylglycerol-3-phosphate-O-acyltransferase 5) in glucagon secretion. Inactivation of *Agpat5* in AgRP neurons of the hypothalamus suppressed their activation by hypoglycemia, impaired hypoglycemia-induced vagal nerve activity, and reduced glucagon secretion. In a hypothalamic cell line and in mice *Agpat5* appeared to divert free fatty acids away from Cpt1a-dependent entry in the mitochondria, FAO, and ATP production. Thus a fall in glycemia leads to decreases in intracellular ATP and AgRP neuron firing. These are interesting and comprehensive studies, and the authors elegantly maneuver between different experimental models to map the details of the involved pathways and mechanisms. The manuscript is clearly written and the data well presented. My only reservation is that the changes in glucagon secretion are rather unimpressive. In addition, one must consider the direct glycemia driven changes in glucagon secretion. So, how much does the hypothalamic mechanism really matter?

Answer: We thank this reviewer for his/her positive comments. Glucagon secretion is controlled by multiple mechanisms, including autonomous nervous activity, the hypothalamic-adrenal axis, intra-islet hormonal interactions, and alpha cells intrinsic sensing mechanisms. Genetically, the control of glucagon secretion is a polygenic trait and, thus, depends on multiple genes, the effect size of each being also dependent on epistatic interactions. Thus, in the genetic screen we performed to identify of hypothalamic regulators of glucagon secretion, it is expected that each of the identified genes exerts only a partial control over glucagon secretion. This reflects the intricacy of the control of hypoglycemia-induced glucagon secretion and our work is aimed at uncovering part of this complexity, as exemplified in our previous studies (ref 5, 36, 37, 38).

I also find it surprising that insulin sensitivity and amounts of glucose needed in clamps are all the same in spite of the reported changes in glucagon responses.

Answer: The hyperinsulinemic-hypoglycemic clamps are performed with high rates of insulin infusion (18 mU/kg/min, now mentioned in the Methods section, page 20, second paragraph) which lead to circulating insulin concentrations that maximally activate the insulin receptor and have a dominant effect in suppressing hepatic glucose production. Thus, relatively small changes in plasma glucagon are not expected to have significant impact on the rate of glucose infusion (GINF).

It is compelling that the *cfos* staining in responses to hypoglycemia depended on *Agpat5* in the AGRP neurons of the ARC as well as the results from patch clamp analysis and Ca⁺ recordings. Vagal activity was found to be increased, but it is not demonstrated that this actually leads to glucagon secretion. One would have expected that also the sympathetic system would be engaged in the response.

Answer: The first response to a fall in glycemia below the euglycemic level is the activation of the vagal nerve (Taborsky and Mundinger, 2012, PMID: 22315452). In a previous study, we showed that activation of the vagal nerve by Glut2 GI neurons of the nucleus tractus solitarius, which activate neurons of the dorsal motor nucleus of the vagus, induces a robust secretion of glucagon (Lamy et al, 2014, Ref 38). Thus, although formally not demonstrated here, we believe that activation of the vagal nerve by hypoglycemia in the present study is also responsible for glucagon secretion. But we have not recorded sympathetic nerve activity here and we cannot exclude that it is also activated.

The case is made regarding mechanisms that the role of *Agpat5* is to divert fatty-acyl-CoAs from Cpt1-mediated entry into mitochondria and this was supported by inactivation of *cpt1a* which restored normal responsiveness. (but again the glucagon levels were not widely different).

Answer: The *Cpt1a* KO mice used to create double *Agpat5* and *Cpt1a* KO mice came from a different breeding facility and were originally on C57BL6/J and not C57BL6/N background. In the experiments performed to assess the impact of *Cpt1a* inactivation on glucose sensing in control or *AgRP^{Agpat5KO}* mice (Figure 6) we studied littermates generated as part of the breeding scheme to obtain double knockout mice. These mice were, thus, of slightly different genetic backgrounds than the Control and *AgRP^{Agpat5KO}* mice studied in rest of the paper. Nevertheless, the c-Fos and electrophysiological experiments clearly supported the hypothesized role of *Cpt1a* in hypoglycemia sensing. The fact that the glucagon response was only weakly decreased in the *Agpat5* KO mice in this experiment probably results from the difference in genetic background and indicates that in these mice the AgRP neurons play a less important role in glucagon secretion. This point is now discussed, Discussion, page 13, first paragraph: “It is interesting to note that in the experiments aimed at testing the effect of *Cpt1a* inactivating in *AgRP^{Agpat5KO}* mice the glucagon response to hypoglycemia was only mildly reduced in the *AgRP^{Agpat5KO}* mice. This could be explained by the fact that the *Cpt1a^{lox/lox}* mice came from a different facility and were on a C57BL6/J and not C57BL6/N background. As all the mice used for this experiment were littermates their genetic backgrounds may have been slightly different from the Control and *AgRP^{Agpat5KO}* mice used in the rest of the study. This indicates that the role of *Agpat5* in AgRP neurons on glucagon secretion depends on epistatic interactions, in line with the results of the genetic screen performed in BXD mice.”

I. 40 GI (neurons) is used as an abbreviation for “glucose inhibited” ; this may be confusing since it often means “gastro-intestinal”

Answer: We understand the point of the reviewer. However, in the field of central glucose sensing and electrophysiology, “GI” is routinely used to denominate glucose-inhibited neurons.

See, for example, the review by Vanessa Routh, 2002 (PMID 12117577). This definition is also mentioned in the Introduction, page 3, second sentence.

I. 43. In humans the main stress induced inhibitor of insulin secretion is the noradrenergic innervation of the pancreas (not the hormones) and probably also the stimulation of glucagon secretion .

Answer: We appreciate the reviewer's observation. Indeed, the noradrenergic innervation of the pancreas plays an important role in glucose homeostasis, in particular in counter-regulatory response and the inhibition of insulin secretion. It also participates in stimulating glucagon secretion, but probably when hypoglycemia falls to levels lower than those activating the vagal nerve (Taborsky and Munding, 2012, PMID: 22315452).

Reviewer #3 (Remarks to the Author):

This paper describes a role of *Agpat5* in AgRP neurons in sensing insulin-induced hypoglycemia and related elevated glucagon secretion. In my view, the overall message is novel and important. I find many of the studies well designed, executed and interpreted. However, there are several results that are incomplete or do not address the critical questions. 1) The paper is about the counter regulatory response of exogenous insulin-induced hypoglycemia. This should be specified in the title. There is hypoglycemia associated with fasting, which is not the focus of the paper. Or if it is the intent of the authors to include all types of shifts in glucose levels in the hypothalamus, they should specify], which experiments is related to which type of hypoglycemia.

Answer: We agree with the reviewer that *Agpat5* was first identified in a genetic screen of glucagon secretion stimulated by insulin-induced hypoglycemia and that our present study focuses of the role of *Agpat5* in AgRP neurons in IIH. We have thus changed the title to: "*Agpat5* in AgRP neurons is required for insulin-induced hypoglycemia sensing and glucagon secretion"

2) The CPT1a work is not consistent with the way these neurons function under fasting conditions, which is the reason I suggest that the authors clearly specify that their focus is insulin-triggered hypoglycemia.

Answer: See answer to point above.

3) The analysis of mitochondrial parameters in the AgRP-specific *Agpat5* KO animals is incomplete, and as it stand, the data is not informative. They should have looked at mitochondria in these animals in the various experimental conditions. On the other hand, these studies are not critical for the main focus and could be dropped.

Answer: We looked at mitochondrial dynamics because previous studies described that phosphatidic acid (PA), which is produced by *Agpat5*, is an inhibitor of Drp1 and reduces mitochondrial fragmentation (Adachi et al., Ref 16, 17). Also, previous studies suggested that Drp1-dependent mitochondrial fragmentation in both POMC (Santoro *et al.* 2017, Ref 25) and AgRP neurons (Dietrich *et al.* 2013, Ref 18; Jin *et al.* 2021, Ref 19) was controlling the activation of these neurons. Here, we looked at the mitochondrial morphology in AgRP neurons only in the fed state when mitochondrial fragmentation is low in control mice; upon *Agpat5* inactivation, we expected that the reduced production of PA should increase Drp1 activity leading to increased mitochondrial fragmentation. This was not observed, indicating that *Agpat5* role in AgRP neurons is not in the control mitochondrial dynamics, at least in the conditions that led to the identification of this gene as regulator of glucagon secretion. Indeed, hypothalamic RNA sequencing, which was part of the initial genetic screen that identified *Agpat5* as well as the present *ex vivo* electrophysiology studies, which revealed difference in glucose sensing between control and *Agpat5* KO neurons, were performed with animals in fed state. Therefore, for the above reasons, we would prefer to keep the data on mitochondrial fragmentation in our manuscript.

4) References 10 and 11 are not supportive of the declarative statement regarding how AgRP neurons are activated during hypoglycemia. Similar to the point above, these are not pertinent to the main finding of the paper.

Answer: The paper by Silver & Erecinska (Ref10) reports seminal observations on the role of the Na⁺/K⁺ATPase in the activation of neurons by hypoglycemia. The paper by Kurita (Ref 11) describes that firing of AgRP GI neurons in response to decrease in extracellular glucose concentrations is triggered by a fall in intracellular ATP content that reduces the activity of the Na⁺/K⁺ATPase. Both papers describe the same essential role of a fall in intracellular ATP and a decrease in the activity of the sodium pump as a critical mechanism inducing GI (and in particular AgRP) neurons firing. In our paper, we refer several times to this mechanism and how *Agpat5* preserves its function. Thus, we respectfully disagree with this reviewer on this point and think that mentioning these references is essential to our paper.

5) Finally, if the paper is about insulin-induced hypoglycemia, which is clearly a crucial medical issue, it would be nice to see how this mechanism is relevant in paradigms when the counter-regulatory response fails.

Answer: We fully agree with this comment, in particular since the pathophysiological mechanisms of iatrogenic hypoglycemia leading to defective counterregulation in insulin-treated individuals with diabetes remain poorly understood. It is the purpose of our current studies to uncover some of the mechanisms responsible for hypoglycemia-induced glucagon secretion in order to pave the way to future investigations on the patho-mechanisms of hypoglycemia-associated autonomic failure. We are presently developing a mouse model of diabetes with repeated hypoglycemic episodes leading to defective counterregulation. These are, however, long-term experiments and are not yet ready for testing all the genes/mechanisms identified in our genetic screen.

REVIEWERS' COMMENTS

Reviewer #1 (Remarks to the Author):

The changes and additions that the authors have made to this revised version of their manuscript are satisfactory and have improved the clarity of their results and conclusions. My congratulations on a very good manuscript and important addition to the field.

Reviewer #2 (Remarks to the Author):

no more comments

Reviewer #3 (Remarks to the Author):

The authors addressed my comments and that of the other reviewers to my satisfaction. It is an excellent paper.